# Diurnal versus spatial variability of greenhouse gas emissions from an anthropogenic modified German lowland river

Matthias Koschorreck[1], Norbert Kamjunke[2], Uta Koedel[3], Michael Rode[4], Claudia Schuetze[3], Ingeborg Bussmann[5]

[1]Department Lake Research, Helmholtz Centre for Environmental Research, Magdeburg, Germany
[2]Department River Ecology, Helmholtz Centre for Environmental Research, Magdeburg, Germany
[3]Department Monitoring & Exploration Technologies, Helmholtz Centre for Environmental Research, Leipzig, Germany
[4]Department Aquatic Ecosystem Analysis, Helmholtz Centre for Environmental Research, Magdeburg, Germany
[5]Department Shelf Sea System Ecology, Alfred Wegener Institut, Helmholtz Zentrum für Polar- und Meeresforschung, Helgoland, Germany

*Correspondence to*: Matthias Koschorreck (matthias.koschorreck@ufz.de)

**Abstract.**

Greenhous gas (GHG) emissions from rivers are globally relevant, but quantification of these emissions comes with considerable uncertainty. Quantification of ecosystem scale emissions is challenged by both spatial and short-term temporal variability. We measured spatio-temporal variability of $CO_2$ and $CH_4$ fluxes from a 1 km long reach of the German lowland river Elbe over three days in order to establish which factor is more relevant to be taken into consideration: small-scale spatial variability or short-term temporal variability of $CO_2$ and $CH_4$ fluxes.

GHG emissions from the river reach studied were dominated by $CO_2$ and 90 % of total emissions was from the water surface, while 10% of emissions was from dry fallen sediment at the side of the river. Aquatic $CO_2$ fluxes were similar at different habitats, while aquatic $CH_4$ fluxes were higher at the side of the river. Artificial structures to improve navigability (groynes) created still water areas with elevated $CH_4$ fluxes and lower $CO_2$ fluxes. $CO_2$ fluxes exhibited a clear diurnal pattern, but the exact shape and timing of this pattern differed between habitats. In contrast, $CH_4$ fluxes did not change diurnally. Our data confirm our hypothesis that spatial variability is especially important for $CH_4$ while diurnal variability is more relevant for $CO_2$ emissions from our study reach of River Elbe in summer. Continuous measurements or at least sampling at different times of the day are most likely necessary for reliable quantification of river GHG emissions.

## 1 Introduction

### 1.1 Greenhouse gas emissions from rivers

Rivers are a globally relevant source of greenhouse gases (GHG) (Battin et al., 2023; Raymond et al., 2012; Rocher-Ros et al., 2023; Stanley et al., 2023). It is currently estimated that rivers globally emit about 2 Pt $CO_2$ $y^{-1}$ (Liu et al., 2022)  and 30.5 ±

17.1Tg $CH_4$ y$^{-1}$ (Rosentreter et al., 2021). However, these estimates suffer from considerable uncertainty. Aquatic GHG emissions actually generate considerable uncertainty among global GHG assessments (IPCC, 2021). Reducing uncertainty in aquatic GHG budgets is important in order to improve biogeochemical models and climate prediction(s). Uncertainties result from a general lack of data as well as from the methods used for budgeting aquatic GHG emissions.

## 1.2 Traditional method for budgeting


Bottom-up approaches to quantifying riverine GHG fluxes are typically based on GHG concentrations measured in a restricted number of water samples. GHG fluxes between water and atmosphere (J) are calculated from concentrations ($C_{water}$) measured in water samples or calculated from other parameters of the carbonate system (pH, alkalinity, and/or DIC) and estimated gas transfer velocities (k) (Raymond et al., 2013) multiplied by water surface area (A) (Equ 1):

$$J = k \; x(C_{water} - C_{atm}) \times A \quad [\text{mol h}^{-1}] \hspace{4cm} \text{Equ. 1}$$

were $C_{atm}$ is the concentration in water which is in equilibrium with the atmosphere. The gas transfer velocity is a physical parameter describing diffusive gas exchange at the water surface and typically estimated from hydrodynamic parameters like flow velocity, slope and/or bottom roughness (Raymond and Cole, 2001). Typical datasets contain weekly to monthly concentration data from a small number of sites along a specific river (Stanley et al., 2023). GHG fluxes can also directly be

measured by floating chambers. However, these measurements are laborious and prone to experimental artefacts if not carried out carefully (Lorke et al., 2015). Water surface areas are typically estimated by using river width based on empirical relations (Raymond et al., 2013) or for larger rivers using remote sensing data (Palmer and Ruhi, 2018). These approaches are suitable to cover seasonal dynamics as well as large-scale spatial patterns along larger rivers. However, recent research has indicated considerable short-term temporal and small-scale spatial variability of riverine GHG fluxes which pose challenges for

budgeting and upscaling.

## 1.3 Short-term temporal variability

There is contrasting evidence for the occurrence of diurnal fluctuation of $CO_2$ in larger rivers (Ishaque, 1973; Haque et al., 2022). The advent of reliable and affordable probes to continuously measure GHG concentrations revealed considerable diurnal fluctuations of $CO_2$ in streams (Gómez-Gener et al., 2021). Because the balance between photosynthesis and respiration

depends on light, $CO_2$ fluxes are typically elevated in the night (Attermeyer et al., 2021). Thus, $CO_2$ emission estimates only relying on discrete water samples taken during daylight hours often significantly under-estimate true emissions.

While diurnal fluctuations of $CO_2$ are well documented, there is little knowledge concerning the short-term variability of $CH_4$ (Stanley et al., 2016).

### 1.4 Small scale spatial variability

While there are several studies investigating spatial variability in streams, much less is known about spatial variability of GHG emissions from larger rivers. Rivers contain various habitat types: these are either natural or those with anthropogenic modifications. Channelisation and disconnection of rivers from their floodplain decrease spatial heterogeneity (Wohl and Iskin, 2019) and most likely also affect GHG fluxes (Machado dos Santos Pinto et al., 2020). However, anthropogenic modifications not only reduce habitat diversity. Several European rivers were modified by building groynes in the 19th century with the

primary goal of concentrating the water into the main river and to improve river flow and navigability (Pusch and Fischer, 2006). Consequently, the flow velocity is lower within the groyne fields leading to increased sedimentation (Kleinwächter et al., 2017). Thus, groynes increase habitat diversity compared to straight, channelised rivers but decrease it compared to a natural shoreline. In the River Elbe, there are no groynes until German river km 120; however, they dominate the shore line downstream from there (Fig. 1b). A recent study presents evidence that the still water areas between such groynes are a source

of $CH_4$ resulting in lateral $CH_4$ gradients (Bussmann et al., 2022). Ignoring these gradients significantly under-estimates total $CH_4$ emissions. The sediment is the predominant source of $CH_4$ in streams (Stanley et al., 2016) and spatial variability of $CH_4$ production in rivers is known to be controlled by sediment deposition (Maeck et al., 2013).

A typical feature of rivers is their fluctuating discharge resulting in fluctuating water level. Thus, depending on discharge, certain parts of rivers are temporarily drying up. It has been shown that these dry river areas emit disproportionally high

amounts of $CO_2$ into the atmosphere (Gómez-Gener et al., 2015). Ignoring dry areas in river GHG budgets may lead to a significant underestimation of GHG emissions (Marcé et al., 2019). Depending on their elevation, such dry river sediments can be quite heterogeneous with respect either to substrate type (sand versus mud), or to the occurrence of temporary vegetation (Bolpagni et al., 2019). Recent research indicates considerable spatial variability as well as temporal dynamics of dry river GHG fluxes (Mallast et al., 2020; Koschorreck et al., 2022).

### 1.5 Aim of this study

We expected that small-scale spatial and diurnal variability would need to be considered for budgeting GHG emissions from rivers. The ideal approach would thus be to perform high-frequency measurements at a large number of sites. However, this is simply not possible and thus, there is a trade-off between frequency and spatial coverage of measurements. In this study, we aim to answer the question "What is more important for budgeting/upscaling GHG emissions from rivers: small-scale spatial

or short-term temporal variability?". We hypothesise that the answer to this question depends on the gas: We expected spatial heterogeneity to be more relevant for $CH_4$ fluxes, while temporal variability is more relevant for $CO_2$. To test this hypothesis, we measured $CH_4$ and $CO_2$ fluxes in different habitats within a typical reach of the German lowland river Elbe over a three-day campaign. The study was designed to cover a typical low-discharge summer situation when both habitat diversity and biological activity in the river were expected to be the highest.

## 2 Materials and Methods

### 2.1 Study Site

Investigations were performed at an one km-long reach of the 8th order River Elbe at Tangermünde located in the middle part of the river in Germany at river km 388 according to German kilometration (Figure 1a). The reach is typical for the middle Elbe River which is characterised by groyne fields (Figure 1b) between km 120-580. Measurements were done in late summer on 18-22 August 2022. Discharge varied between 189-197 $m^3\ s^{-1}$ during that period which was below the mean low discharge of 235 $m^3\ s^{-1}$ (Figure 2).

We separated the river's water surface into three distinct habitats: The middle of the river, the sides of the river, and the area between groynes (groyne fields). The groyne fields extend from the riverbank to a virtual line connecting the heads of the groynes. The side areas we defined as extending from the outer boundary of the groyne fields 15 m into the river (about 10% of river width). Visual inspection confirmed that 15m fully included the turbulent areas below the groyne heads. The studied reach had ten groynes at both banks, extending up to 60 m into the river. The distance between the groynes was $80 \pm 10$ m. The area between the groynes was partly dry. These dry areas featured three typical habitats: muddy areas and sandy beaches without and with terrestrial vegetation (Figure 1c, Figure S1).

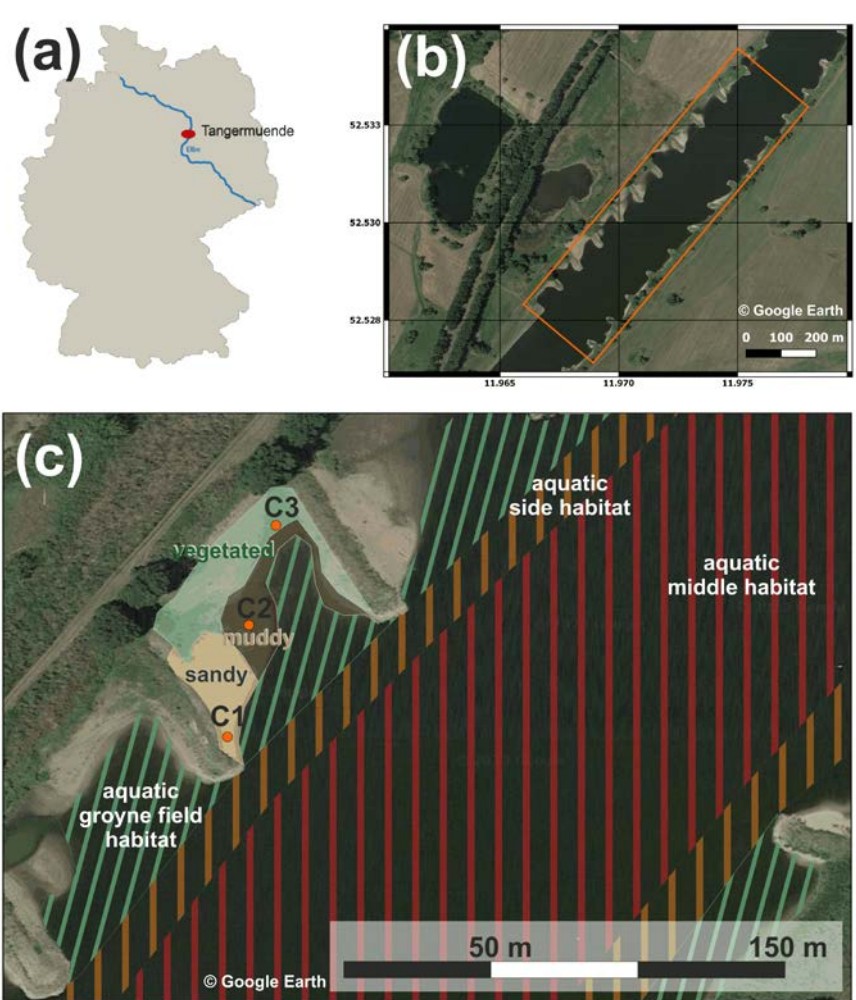

**Figure 1: Location of the investigation site Tangermünde (red dot) within Germany at River Elbe at river km 388 according to German kilometration (a). Top view of the sampling reach with groyne fields (Source orthophoto: Google Earth, GeoBasis-DE/BKG, date of recording 06/08/2020), flow direction to North-east, study area marked by orange lines (b). Detailed view of the study area with indication of habitat types (c). The groyne field is divided into an aquatic habitat and the terrestrial habitat with partially dry fallen sediments, which are divided into sandy areas (location of soil flux chamber C1), muddy areas (chamber C2) and vegetated areas (chamber C3). (Source orthophoto: Google Earth, GeoBasis-DE/BKG, recording date 06/08/2020, © Google Earth, see also Fig. S1).**

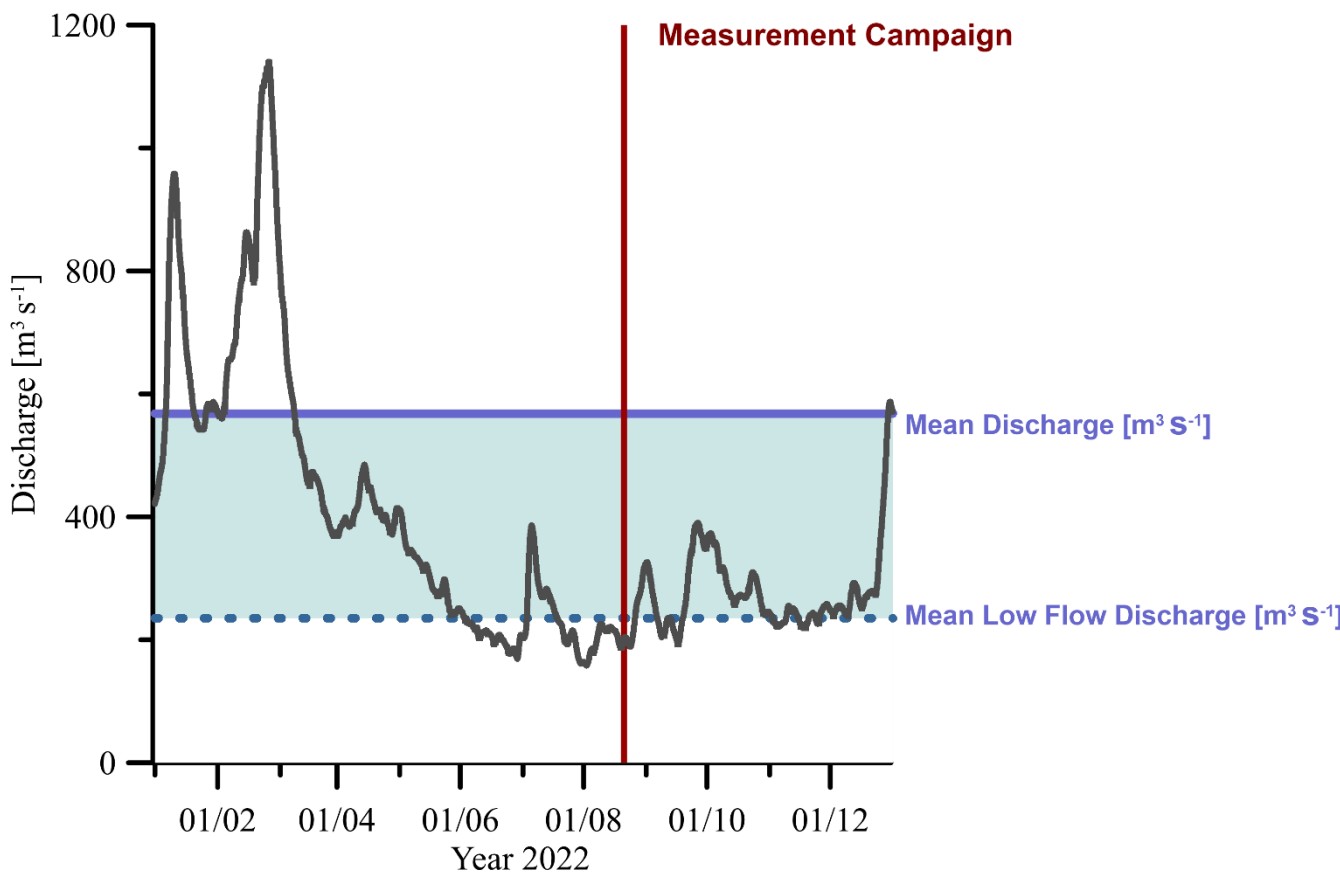

**Figure 2: Discharge of River Elbe at gauge Tangermünde during 2022 (German Federal Waterways and Shipping Administration). The blue area marks the range between mean low flow discharge (annual mean of lowest discharge of each month) and mean discharge, the red area marks the period of the sampling campaign.**

## 2.2 Aquatic Measurements

Table 1 provides an overview over the different methods used to assess GHG concentrations and fluxes.

### 2.2.1 Hydrodynamics and basic physicochemical measurements

Flow velocity profile measurements were conducted at the study site at Tangermünde on 19 August 2022. We deployed a Teledyne RD Instruments (TRDI) four-beam 1200 kHz Rio Grande ADCP™ from an inflatable boat. The vertical resolution of water velocities was 0.25 m (25 cm bins), and the sampling frequency was approximately 2 Hz. In total 8 transects of flow velocity and water depth where measured using 4-5 replicates at each transect (Figure S3 in the Supplement). Depth-averaged velocities ($U$) were calculated from the measured portion of the water column, neglecting the unmeasured upper and lower portion of the water column (see Figure S3 in the Supplement). Global positioning system (GPS) position data were collected using a GPS tracker (Garmin International) with a frequency of 1s.

Basic hydrographic parameters (temperature, conductivity, oxygen, pH, turbidity and chlorophyll) were determined with a pocketferrybox (4Hjena, Kiel, Germany) and a multiparameter probe (Exo2, YSI). The water supply for both sensors was the ship's duct with direct water supply of the RV Albis. We mapped basic hydrographic parameters with the RV Albis meandering between the western and eastern groyne heads on 19 August 2022. Due to the size of the ship, it was not possible to enter near-shore areas and groyne fields. From 19-22 August 2022, the RV Albis was anchored at the pier in Tangermünde and measured the hydrographic parameters at the same position, continuously for 63 hours.

### 2.2.2 Aquatic GHG measurements

To assess the spatial pattern of aquatic GHG fluxes, aquatic GHG fluxes were measured from an inflatable boat using a floating chamber connected to a portable FTIR analyser (GASMET) as explained in Lorke et al. (2015). We used exactly the same rectangular drifting chamber (area 0.098 $m^2$, height 0.15 m) as at River Bode in Lorke et al. (2015). For each measurement the chamber was deployed from a drifting boat for 2-5 minutes. Fluxes were calculated from the linear change of $CO_2$ and $CH_4$ mixing ratios in the chamber. A water sample for later gas chromatograph (GC) analysis was taken during each flux measurement. Surface water samples of 30 mL were taken with 60 mL syringes, and 30 ml of ambient air was added to the syringes, which was then vigorously shaken for one minute. The equilibrated headspace was then transferred to pre-evacuated exetainers and the equilibration temperature was measured in the remaining water sample.

Dissolved methane concentrations were mapped with a dissolved gas extraction unit and a laser-based analytical Greenhouse Gas Analyzer (GGA; both Los Gatos Research, United States) on an inflatable boat. The degassing unit withdrew water from the water basin or directly from surface water at 1.2 L min $^{-1}$. Methane was extracted from the water via a hydrophobic membrane and hydrocarbon-free carrier gas was on the other side of the membrane (nitrogen, at 0.5 L min $^{-1}$). The carrier gas with the extracted $CH_4$ was then directed to the inlet of the gas analyser. The time offset between the water intake and stable recording at the GGA was determined beforehand in the laboratory. To convert the relative concentrations (ppm) given by the GGA to absolute concentrations (nmol $L^{-1}$), discrete water samples were obtained at least every hour. The $CH_4$ concentration in these bottles was determined using the headspace method and gas chromatographic analysis. The range of concentrations from the water samples used for calibration was rather narrow (178 – 258 nmol/L), thus we used a conversion factor (water sample conc. / ppm from GGA) which was 88.7 ± 23 nM / ppm) (Table S1).

For high-spatial resolution measurements the degassing unit and GGA were set-up in a small rubber boat with a 5-L nitrogen tank and a car battery. The water inlet of the degasser was fixed to a bar and submerged to approx. 20 cm water depth. We entered each groyne field from the north and kept the boat at the groyne heads for approximately 2 min. (against the current) and then entered the following groyne field as far as possible (Figure 3).

For continuous measurements of dissolved $CO_2$ and $CH_4$, an optical AMT-Sensor (Rostock, Germany) and a Contros-Sensor (4H-Jena, Jena, Germany) were deployed in the ship's duct of the Albis. The $CO_2$ sensor provided data as ppm which were converted to concentrations (µmol $L^{-1}$) according to its solubility at the respective temperature (UNESCO/IHA, 2010). The

$CH_4$ mixing ratio (ppm) values of the Contros sensor were converted to absolute concentrations ($\mu$mol L$^{-1}$) by relating them to water samples measured with a GC, similar to the values from the GGA (LosGatos). The conversion factor here was 0.06 $\mu$mol L$^{-1}$ ppm$^{-1}$. Probes were checked in the laboratory prior to deployment by comparing probe readings with concentrations measured by GC and/or a membrane equilibrator conneted to an NDIR analyser as explained in (Koschorreck et al., 2021).

## 2.3 Terrestrial and atmospheric measurements

The mixing ratio of $CO_2$ in the atmosphere and other meteorological parameters was continuously measured by a sensebox. The sensebox system is a toolkit developed in the framework of Citizen Science projects for environmental data collection funded by the German Federal Ministry of Education and Research. It consists of an open-source microcontroller unit which can deploy various environmental sensors (Bartoscheck et al., 2019). The sensebox was equipped with a GPS-sensor, an environmental sensor measuring air temperature, relative humidity, and air pressure, and a $CO_2$ - sensor determining the air $CO_2$ mixing ratio. The sensebox was installed at the RV Albis and measured the atmospheric conditions every 5 minutes.

The spatial variability of $CO_2$ and $CH_4$ fluxes at terrestrial sediments was measured with the laser-based trace gas analyser LI-7810 (LI-COR Biosciences, USA) in combination with a closed chamber (LI-COR smart chamber). Soil gas fluxes were calculated from the temporal gas concentration change taking into consideration the chamber volume and the surface area of the soil area covered by the chamber. For each sampling point the change in gas concentration in the closed chamber was determined after a purging period in a 1-second sampling interval during the 2-minute observation time. Fluxes were calculated using the linear fitting approach of the SoilFluxPro Software (LI-COR). Based on our own long-term tests with the system, soil flux detection limits were determined for $CO_2$ of $\pm$0.36 mmol m$^{-2}$ h$^{-1}$ (corresponding to a change of 2 ppm $CO_2$ during a closure time period of 2 minutes) and for $CH_4$ of $\pm$0.072 $\mu$mol m$^{-2}$ h$^{-1}$ (corresponding to a change of 0.6 ppb $CH_4$ during a closure time period of 2 minutes). To assess spatial variability we measured GHG fluxes at 5 sandy, 5 muddy, and 9 vegetated sites (Figure S2). Muddy and sandy areas were free from vegetation and could be clearly distinguished from vegetated zones, which were widely covered by typical herbaceous plants such as *Persicaria*, *Inula britannica*, and *Xanthium strumarium*.

To cover the temporal dynamics of terrestrial $CO_2$ fluxes, three opaque automatic chambers (CFLUX-1 Automated Soil $CO_2$ Flux System, PP systems, Amesbury, Massachusetts, USA) were installed at a sandy site, a muddy site, and a sandy site with herbaceous vegetation (Figure 1 d, Figure S2). The chambers measured $CO_2$ fluxes once every hour. Each flux measurement lasted for 5 min, and the chambers were open for 55 min between flux measurements. $CO_2$ fluxes were calculated from the linear increase of $CO_2$. The detection limit for terrestrial $CO_2$ fluxes was 0.08 mmol m$^{-2}$ h$^{-1}$. Reliability of the $CO_2$ measurement in the authmatic chambers was checked by comparing the atmospheric background concentrations measured indepently by the three automatic chambers. Each chamber was equipped with a soil moisture and temperature probe (Stevens HydraProbe, Stevens Water Monitoring Systems, Portland, Oregon, USA).

Light intensity was measured at Magdeburg (50 km from Tangermünde) as PAR [$\mu$mol m$^{-2}$ s$^{-1}$] using a LI-190R Quantum Sensor (Licor, Lincoln, U.S.A.).

Table 1: Overview of methods used to measure different parameters. Precision is defined as two times the standard deviation of at least 10 consecutive measurements (5 replicate image analyses for area). The precision of the terrestrial flux measurements based on survey chamber investigations is site specific and determined using the fitted linear gas concentration curves.

| | method | precision | when measured |
|---|---|---|---|
| **spatial variability** | | | |
| aquatic $CO_2$ concentration | GC samples | 0.14 µmol $L^{-1}$ | 20.8. 11:25 – 14:10 |
| aquatic $CH_4$ concentration | degasser + GHG analyser | 2 nmol $L^{-1}$ | 21.8. 16:18 – 16:52 |
| aquatic $CO_2$ flux | floating chamber + GHG analyser | 0.5 mmol $m^{-2}$ $h^{-1}$ | 20.8. 11:25 – 14:10 |
| aquatic $CH_4$ flux | floating chamber + GHG analyser | 8 µmol $m^{-2}$ $h^{-1}$ | 20.8. 11:25 – 14:10 |
| terrestrial $CO_2$ flux | survey chamber | 0.5 mmol $m^{-2}$ $h^{-1}$ | 19.8. 12-14, 20.8. 10-14, |
| terrestrial $CH_4$ flux | survey chamber | 0.1 µmol $m^{-2}$ $h^{-1}$ | 21.8. 4:30-5, 7:30-9 |
| $k_{600}$ | calculated from flux and concentration | 0.52 m $d^{-1}$ | 20.8. 11:25 – 14:10 |
| areas | Google Earth images | 200 $m^2$ | 6 images |
| **temporal variability** | | | |
| aquatic $CO_2$ concentration | $CO_2$ probe | 0.1 µmol $L^{-1}$ | 19.8. 16:00 – 22.8. 6:00 |
| aquatic $CH_4$ concentration | $CH_4$ probe | 1 nmol $L^{-1}$ | 19.8. 15:00 – 22.8. 7:00 |
| aquatic $CO_2$ flux | calculated from concentration + $k_{600}$ | 0.06 mmol $m^{-2}$ $h^{-1}$ | 19.8. 16:00 – 22.8. 6:00 |
| aquatic $CH_4$ flux | calculated from concentration + $k_{600}$ | 0.3 µmol $m^{-2}$ $h^{-1}$ | 19.8. 15:00 – 22.8. 7:00 |
| terrestrial $CO_2$ flux | automatic chambers | 2.2 mmol $m^{-2}$ $h^{-1}$ | 18.8. 16:00 – 22.8. 6:00 |
| terrestrial $CH_4$ flux | not measured | | |

## 2.4 Laboratory Analyses

$CO_2$ and $CH_4$ concentrations in gas samples were measured with a gas chromatograph (GC) (SRI 8610C, SRI Instruments Europe, Bad Honnef, Germany). The GC was equipped with a flame ionisation detector and a methanizer which allowed for simultaneous measurement of $CO_2$ and $CH_4$ with an uncertainty of < 5 %. Dissolved gas concentrations were calculated using temperature-dependent Henry coefficients (UNESCO/IHA, 2010). $CO_2$ concentrations were corrected for alkalinity as described in Koschorreck et al. (2021).

## 2.5 Calculations and Statistics

Gas transfer coefficients were calculated from $CH_4$ fluxes measured by the floating chambers divided by the difference between actual and equilibrium $CH_4$ concentration. Equilibrium $CH_4$ concentrations were calculated from the mean measured atmospheric $CH_4$ mixing ratio (2.5 ppm) using temperature-dependent Henry coefficients from Sander (2015); (Koschorreck

et al., 2021). The so-determined $k_{CH4}$ was converted to $k_{600}$ and $k_{CO2}$ using Schmidt numbers according to UNESCO/IHA

(2010). We did not use $CO_2$ data for $k_{600}$ calculations because the $CO_2$ concentration was close to equilibrium resulting in large uncertainties in the calculation of $k_{600}$.

Probe measurements of $CO_2$ and $CH_4$ concentrations measured at RVAlbis were converted to fluxes using the measured gas transfer velocity of $k600 = 5.2$ m d$^{-1}$ (Table 2). This assumes that $k600$ at the probe site was equal to the mean $k600$ measured in the side habitat. For $CO_2$ fluxes we used the measured atmospheric $CO_2$ mixing ratios while for $CH_4$ we used a constant

atmospheric mixing ration of 2.5 ppm. $k600$ was converted to $kCO_2$ and $kCH_4$ as explained above.

Fluxes from different habitat types were compared by pairwise Wilcoxon tests. Fluxes during the day were compared to fluxes during night using Wilcoxon rank sum tests. For statistical analysis, all high-frequency data were transformed to hourly data by calculating the mean values of data available between 30 minutes before and 30 minutes after the full hour. Time series data were log transformed after checking for normality by using Shapiro-Wilk tests. Since we sometimes observed slightly

negative fluxes, fluxes were corrected by adding the most negative flux to all flux data before log-transformation. Significance of linear correlation of log transformed fluxes with drivers was checked by F-tests ($p<0.05$). Mixed linear models to explain GHG fluxes from combinations of predictors were compared based on their AIC (Bates et al., 2015). To consider site-specific correlations, we added site as a random factor to our statistical model. All statistical analyses were done with R (R-Core-Team, 2016).

**3 Results**

**3.1 Hydrodynamic and climatic conditions**

Our study was conducted during a typical summer low-water situation. At the gauge level of 134 cm (on 6 August 2022, when the orthophoto in Figure 1 was taken), 9 % of the study area was not covered by water (Table 2). Flow velocity in the river was relatively uniform around 0.65 m s$^{-1}$ while within the groyne fields flow velocity was significantly lower (Table 2). The

minimum and maximum measured flow velocity were 0.007 and 1.05 ms$^{-1}$, and the minimum and maximum water depths were 0.45 and 3.1 m respectively. Water was flowing rather smoothly without larger waves, but we observed more turbulent conditions downstream of groyne heads. The mean water depth was 1.82 m with the most shallow but also the deepest parts in the groyne fields (maximum depth 3.1 m). The water level rose by 10 cm over the 3 days of our study. Weather conditions were rather constant during our study period, predominantly sunny, with 5.6 mm of rainfall during the first night (19.8. until

6 am). Air temperature fluctuated between 12.4° and 32.2° and low wind speeds of $0.3 \pm 0.6$ m s$^{-1}$ were recorded (Fig.S7). Water temperatures were evenly distributed ($23.3 \pm 0.04$°C, measured between 19.08.2022 14:00 and 22.08.2022 08:00). Electrical conductivity of the water at the western shore and within the western groyne fields was about 200 µS cm$^{-1}$ higher than on the eastern shore (Fig. S4). This conductivity gradient has been attributed to the salty inflow of river Saale 97 km

upstream of our sampling site (Weigold and Baborowski, 2009). This slight difference in conductivity most probably does not

affect microbial GHG production but it indicates limited lateral mixing of river water even over a large distance.

**3.2 Spatial variability**

GHG concentrations and fluxes differed between habitats and also between the two gases $CO_2$ and $CH_4$ (Table 2). The water was over-saturated both with $CO_2$ and $CH_4$, resulting in positive fluxes (= emission to the atmosphere). The flux of $CO_2$ from the water surface was two orders of magnitude higher than the $CH_4$ flux. Dissolved $CO_2$ concentrations were highest in the

middle of the river and decreased towards the side. An opposite pattern was observed for $CH_4$ which was higher at the sides and within the groyne fields than in the middle of the river (Figure 3). Since the gas transfer velocity k600 was twice as high at the sides compared to the middle of the river, $CH_4$ fluxes were significantly higher at the side and in the groyne field compared to the middle of the river. We never observed ebullition in our chamber measurements. Since higher $CO_2$ concentrations in the middle of the river were compensated by lower k600 values, fluxes of $CO_2$ did not differ significantly

between aquatic habitats (Figure 4). Sediment incubations (methods in SI) confirmed that $CH_4$ was mainly produced in the sediment. In sediment samples from a groyne field $CH_4$ was produced with a rate of $2095 \pm 2781$ mol $L^{-1}$ $h^{-1}$. Oxic water samples also produced methane with a low rate of $1.73 \pm 0.5$ mol $L^{-1}$ $h^{-1}$.

The dry habitats had higher $CO_2$ fluxes but lower $CH_4$ fluxes compared to the aquatic sites. Dry $CO_2$ fluxes showed a large

variability within habitats and were highest at vegetated sites and very low at the sandy sites (Figure 4). Dry $CH_4$ fluxes, in contrast, were more than two orders of magnitude lower, in some cases even slightly negative (Table 2), and only small differences between habitats were observed.

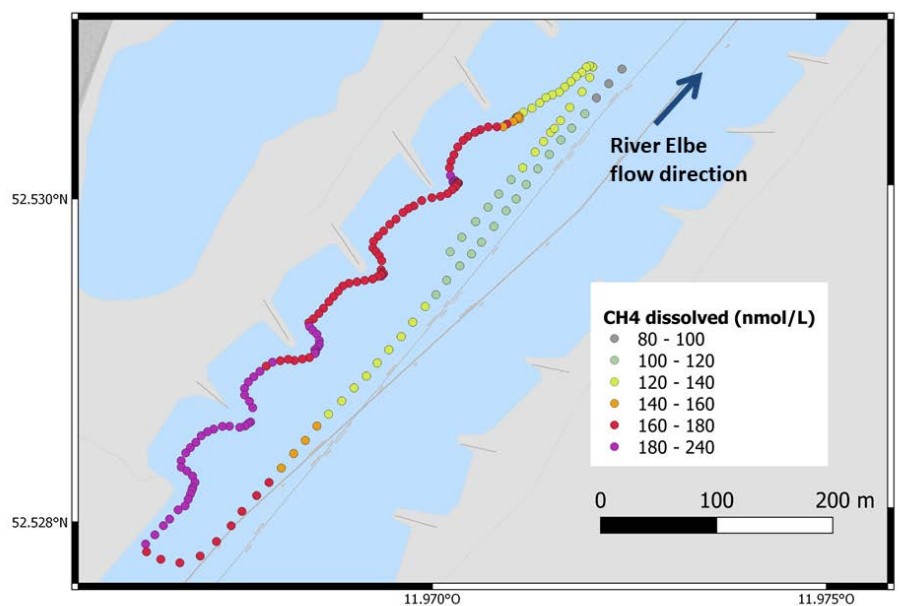

**Figure 3: Concentrations of dissolved CH₄ in the water measured continuously with a mobile gas extraction unit connected to a GHG analyser. Measurements were carried out from an inflatable boat on 21.8.2022 between 16:18 and 16:51. CO₂ was also measured by the GHG analyzer but data were not used because gas extraction was different for CH₄ and CO₂ and the system was optimized for CH₄.**


Table 2: GHG fluxes and their regulating factors in different habitat types. Emission data from floating chamber measurements, $CH_4$ concentration measured by a membrane equilibrator connected to a portable GHG analyser (data from Figure 3), $CO_2$ concentration analysed by GC, velocity measured by ADCP. Areas were manually extracted from Google earth images. all data: median (range). $CH_4$ fluxes are given in $\mu mol\ m^{-2}\ h^{-1}$ and $CO_2$ fluxes in $mmol\ m^{-2}\ h^{-1}$. Original data of terrestrial fluxes are published in Koedel and Schütze (2023).

| | aquatic | | | terrestrial | | |
|---|---|---|---|---|---|---|
| | middle | side | groyne fields | sandy | muddy | vegetated |
| $CH_4$ fluxes [$\mu mol\ m^{-2}\ h^{-1}$] | 12.5 (4-20.8) | 41.7 (21-75) | 39.6 (12.5-54.2) | 0.12 (-0.2-0.38) | 0.03 (-0.83-1.13) | -0.006 (-2.9-0.5) |
| $CO_2$ fluxes [$mmol\ m^{-2}\ h^{-1}$] | 2.2 (1.86-2.21) | 1.5 (1.07-3.13) | 0.8 (0.21-2.3) | 1.9 (0.7-4.6) | 4.9 (-4.1-16.8) | 11.1 (-2.7– 45.1) |
| $CH_4$ concentration [$\mu mol\ L^{-1}$] | 0.12 (0.11 - 0.16) | 0.18 (0.17 - 0.32) | 0.18 (0.17 -0.21) | - | - | - |
| $CO_2$ concentration [$\mu mol\ L^{-1}$] | 29.3 (28.4-29.6) | 23.4 (15.4-24.2) | 15.7 (15.4-28.8) | - | - | - |
| k600 [$m\ d^{-1}$] | 2.6 (1.6-5.2) | 5.2 (2.2-10.3) | 4.3 (1.4-5.7) | - | - | - |
| velocity [$m\ s^{-1}$] | 0.79 (0.72-0.81) | 0.65 (0.22-0.84) | 0.22 (0.09-0.45) | - | - | - |
| total area [$m^2$] | 125,000 (112.000-132,000) | 25,000 (18,000-28,000) | 42,000 (38,600-45,000) | 7,700 (6,000-8,900) | 4,800 (3,500-5,500) | 6,500 (5,600-7,800) |
| total $CH_4$ emissions [$mol\ h^{-1}$] | 1.6 (0.5 - 2.6) | 1.0 (0.2 - 1.8) | 1.6 (0.5 - 2.3) | 0.009 (-0.002-0.003) | 0.0002 (-0.004-0.005) | -0.00004 (-0.019 - 0.003) |
| total $CO_2$ emissions [$mol\ h^{-1}$] | 275 (232 - 276) | 37.5 (26.7 - 78.5) | 33.6 (8.8 - 96.6) | 14.3 (5.5-35.4) | 23.6 (-19.3-80.7) | 72.4 (-17 - 293) |

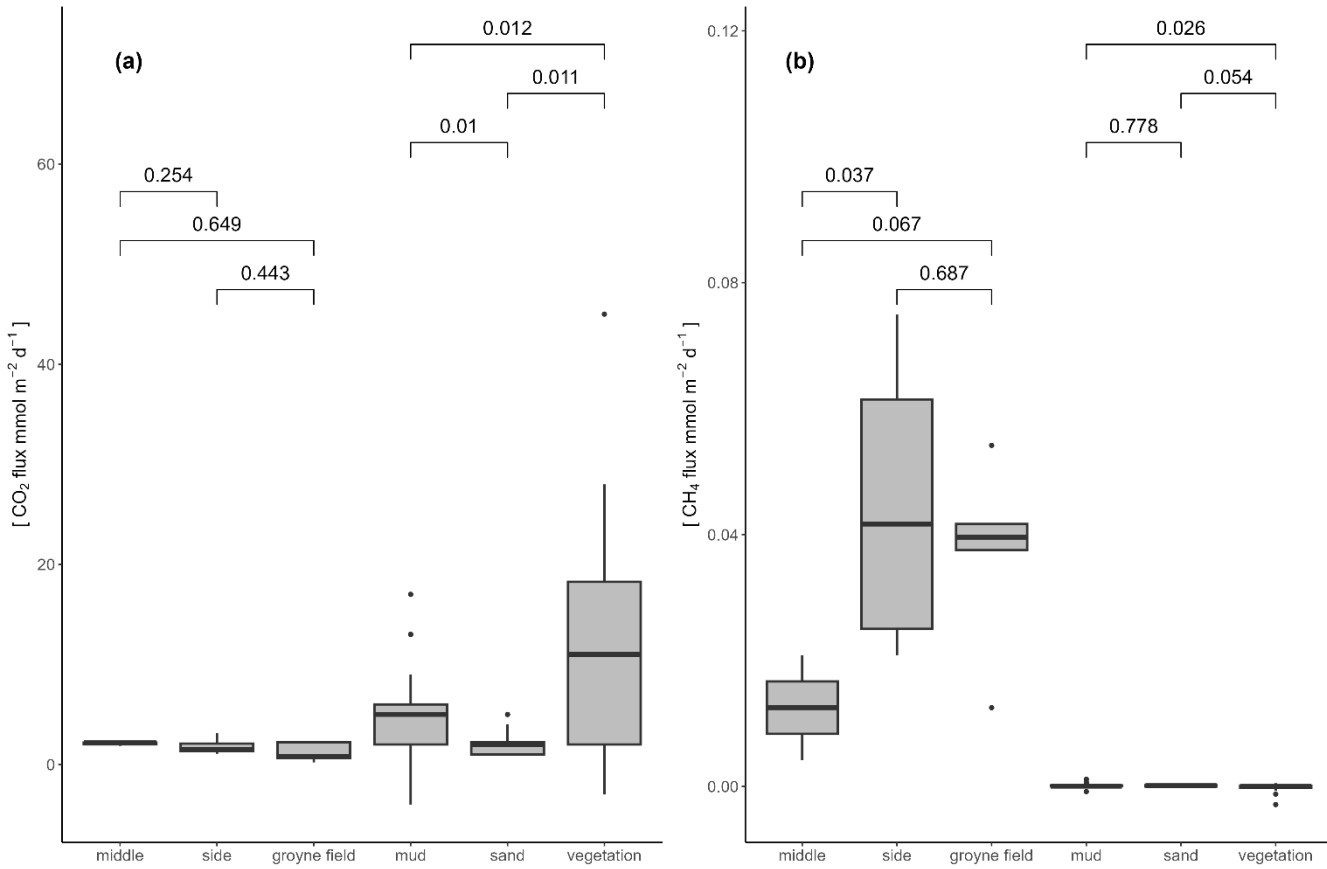

**Figure 4: Boxplots comparing $CO_2$ flux (a) and $CH_4$ flux (b) in different habitat types. Numbers within the plots indicate p-values of pairwise comparisons using Wilcoxon tests (same data as in table 2).**

### 3.3 Temporal variability

Continuous measurements of aquatic GHG concentrations (Figure S5) and water physicochemical variables (Figure S6) were carried out over a period of 2.5 days at the side of the river. Water temperature (median 22.6; range 21.7 - 23.7°C), pH (median 8.0; range 7.3 - 8.3) and oxygen (median 97.9; 87.5 - 117.8 % saturation) showed a clear daily pattern, with lowest values in the early morning (5:00 - 6:00), while conductivity was rather constant (median 1260; range 1146 - 1381 µS/cm; Supp. Figure S6).

Aquatic $CO_2$ concentrations showed a clear diurnal cycle with rising concentrations during the night and decreasing concentrations during the day. The diurnal amplitude of the resulting flux spanned about 5 mmol $m^2$ $h^{-1}$ with maximum fluxes around 5:00 and minimum fluxes around 19:00 (blue line in Figure 5a). In contrast to $CO_2$, the $CH_4$ concentration and the flux did not change with time (Figure 5b).

At the dry sites diurnal pattern of $CO_2$ fluxes were apparent, but these patterns differed between habitat type. At the vegetated site a pattern similar to the aquatic site was observed, but phase shifted. The highest fluxes were found around noon, while the minimum was before sunrise (green line in Figure 5a). The mud site did not show such a sinus-like pattern but rather a two-state pattern: The site switched from constant fluxes during the night to constant $CO_2$ uptake during the day (red line in Figure 5a). No diurnal variability of $CO_2$ fluxes was observed at the sandy site (yellow line in Figure 5a). The $CH_4$ flux data from the dry sites do not allow visualisation of diurnal variability, but the data (dots in Figure 5b) at least do not show any temporal trend during the day.

$CO_2$ fluxes from the mud and the vegetated site showed a decreasing trend during our study. At the muddy site this trend was associated with increasing sediment moisture (Figure 5c), which was obviously caused by the rising water level of the river (Figure 5d) - on 21.8. the flux chamber was only 1 m from the water line. At the other dry sites no trend of sediment moisture was visible, but sediment temperature during the day tended to increase through the study (Figure 5c). There was light rain during the first night (Figure 5d) which resulted in a slight increase of sediment moisture as well as $CO_2$ flux only at the vegetated site.

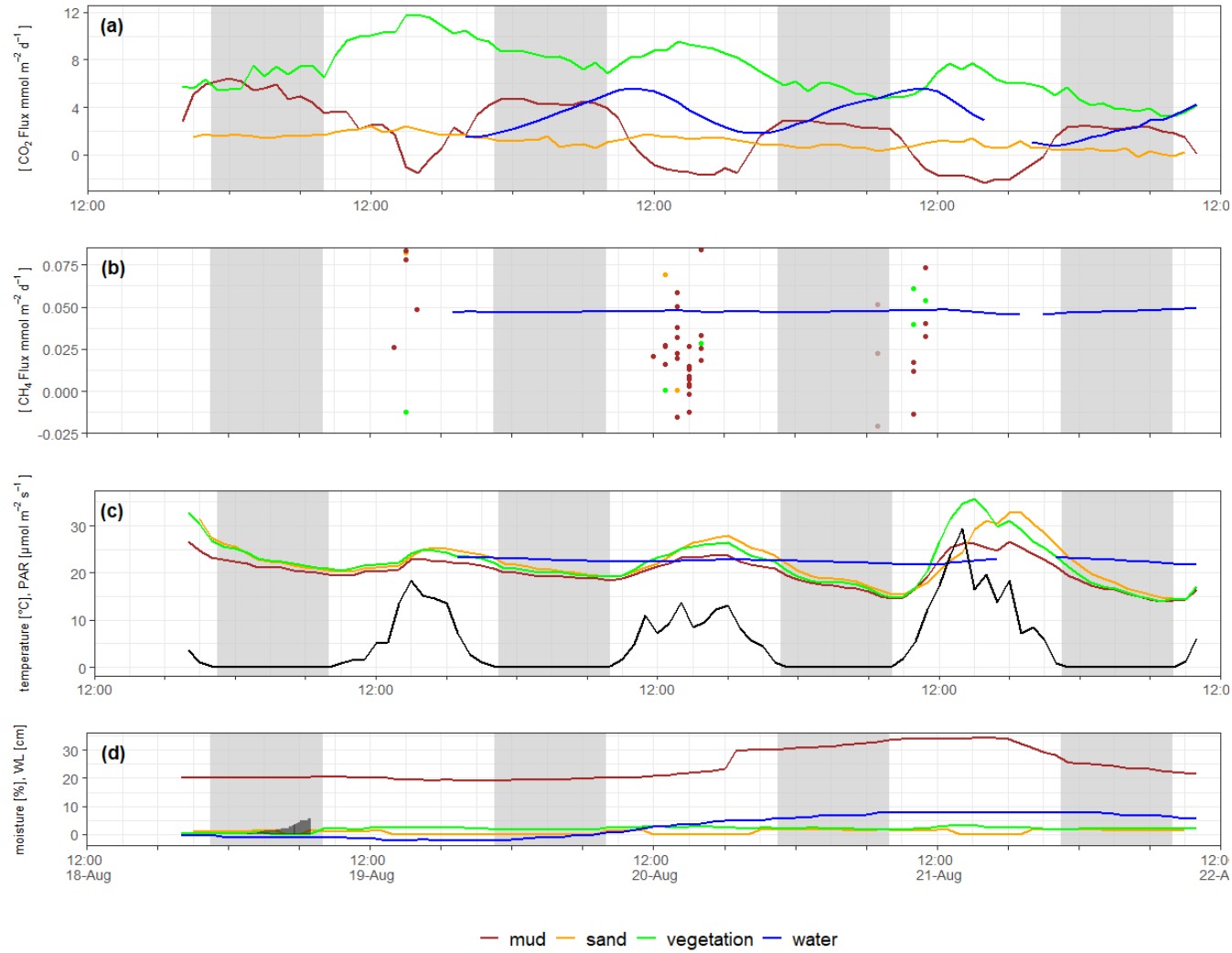

**Figure 5: Time series of CO₂ flux (a), CH₄ flux (terrestrial CH₄ fluxes shown as dots since they were measured at different spots) (b), sediment and water temperature (c), and sediment moisture and water level (d). All data with hourly resolution. Black line in c) shows light measured as PAR [μmol m⁻² s⁻¹/2]. Grey-shaded areas indicate night. Grey bars in (d) indicate cumulative rain [mm] in the first night. Time is UTC. Please note that time series data were measured indepently from spatial data in table 2.**

As a result of these diurnal patterns, fluxes were significantly higher during the day than during the night at the sandy and vegetated site, while at the muddy site fluxes were higher at night than during daylight (Table 3). At the water site, medians did not differ significantly between day and night.

We correlated the dry $CO_2$ flux with measured potential drivers (Figure S8). The $CO_2$ flux was weakly negatively correlated with sediment moisture and water level. Correlation with temperature or light (which were significantly linearly correlated, F-test $p < 0.05$) including all data was not significant (F-test, $p > 0.05$), which is consistent with the observed different diurnal pattern at different sites. Furthermore, the temperature of the water was relatively constant – in contrast to the dry sites where

diurnal temperature amplitudes up to 20° were observed (Figure 5c). If we correlate the $CO_2$ flux with temperature or light for
each habitat type separately, we get a significant positive correlation at the sandy and vegetated site, while the correlation was
negative at the muddy and aquatic site (Figure S9, Table S2). A positive correlation with light was observed at the vegetated
site, while the correlation was negative at the muddy site.

A mixed linear model with (log-transformed) temperature, light, water level, and sediment moisture as fixed factors and site
as random factor explained 76 % of the variability. The most parsimonious model (based on AIC) contained water level as
fixed factor and had a conditional $R^2$ of 0.75 (Table S3).

Table 3: Median (range) of temporal GHG fluxes at different sites (data from Figure 5). Day and night separated by sunrise/sunset.
Day and night data were significantly different for the dry sites but not for the water (Wilcox test).

|  | habitat | all data | day | night |
|---|---|---|---|---|
| $CO_2$ flux | mud | 2.3 (-2.34 - 6.41) | 0.07 (-2.34 – 5.08) | 3.56 (1.51 - 6.41) |
|  | sand | 1.19 (-0.22 - 2.45) | 1.37 (-0.11 - 2.45) | 0.81 (-0.22 - 1.76) |
|  | vegetation | 7.06 (3.24 - 11.84) | 7.7 (3.24 - 11.84) | 6.01 (3.24 – 9.58) |
|  | water | 3.32 (0.80 - 5.96) | 3.78 (0.90 - 5.64) | 2.67 (0.76 – 4.8) |
| $CH_4$ flux | water | 0.052 (0.05 - 0.055) | 0.047 (0.045 - 0.049) | 0.047 (0.046 - 0.048) |

**3.4 Comparison of spatial and temporal variability**

We calculated coefficients of variation (CV) to make spatial and temporal variability comparable (Figure 6). Consistent with
our hypothesis, spatial variability of $CH_4$ fluxes had a higher CV compared to $CO_2$ in  dry habitats. Also consistent with our
hypothesis, temporal variability of the $CO_2$ flux had a higher CV than $CH_4$ flux at the aquatic sites. Spatial variability of both
gases was similar in aquatic habitats (CV = 0.5). At the dry sites (where we did not measure temporal changes in $CH_4$ flux)
the temporal variability of the $CO_2$ flux also showed a high CV. If we want to judge the consequences of this result on
upscaling, however, the absolute height of the fluxes as well as the relative areas of the different habitats need to be considered.

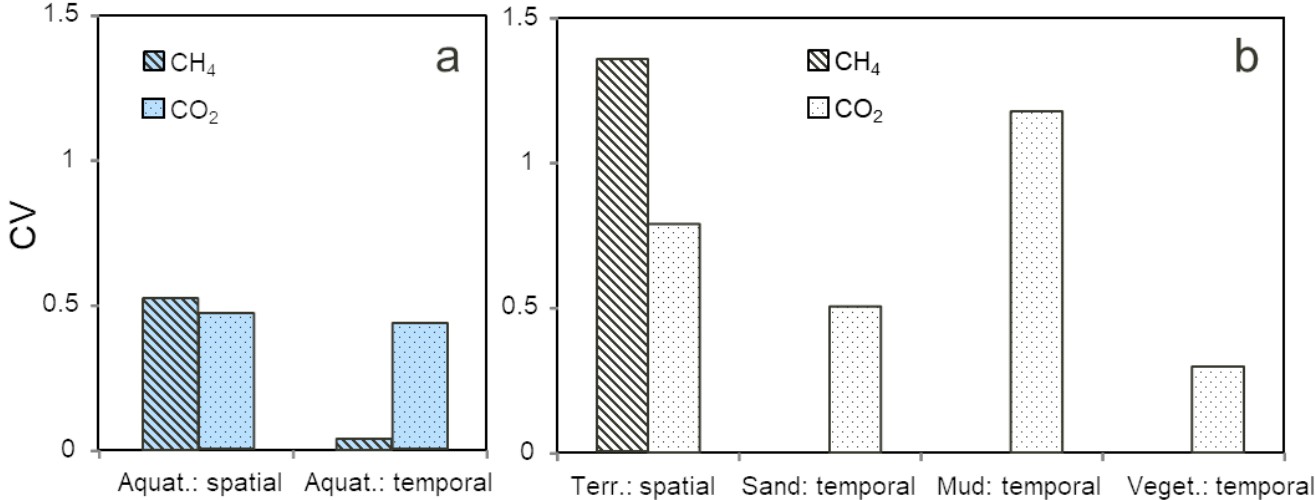

**Figure 6: Coefficients of variation of spatial (calculated from Table 2) and temporal (calculated from Figure 5) variability of CH₄ fluxes and CO₂ fluxes of aquatic sites (a) and dry sites (b) accordingly. Temporal variability of CH₄ fluxes from terrestrial sites was not calculated.**

### 3.5 Upscaling

About 10 % of our study area was not covered by water (Table 2). Within the dry area, all three habitat types contributed similarly to the total area. If we apply an "optimal" approach considering spatial variability of $CH_4$ and both spatial and temporal variability for $CO_2$, our study reach emitted 91 mol $CH_4$ d$^{-1}$ and 16962 mol $CO_2$ d$^{-1}$ (Table 4). Thus, $CH_4$ contributed 5 % to total $CO_2$-eq emissions. GHG emissions were dominated by the aquatic habitats which contributed 91 % to the total emissions. $CH_4$ emissions from terrestrial habits can be neglected, while at the aquatic habitats they contributed about 6 % to the $CO_2$-eq emissions.

**Table 4: Total GHG emissions from our studied reach quantified considering both spatial and temporal variability. For CO₂ emissions we multiplied the median flux from the temporal data (Figure 5) by habitat areas (Table 2). For aquatic CO₂ emissions, the temporal median flux at the side was also applied to the other aquatic habitats. For CH₄ emissions we used the spatially different fluxes (Table 2). For CO₂-eq the CH₄ emissions were converted to CO₂-eq using a GWP of 28 and added to the CO₂ emissions.**

|  | aquatic | | | | terrestrial | | | | all |
|---|---|---|---|---|---|---|---|---|---|
|  | middle | side | groyne fields | total | sandy | muddy | vegetated | total | total |
| $CH_4$ [mol d$^{-1}$] | 37.5 | 25.02 | 40 | 102 | 0.006 | 0.014 | -0.001 | 0 | 91 |
| $CO_2$ [mol d$^{-1}$] | 9960 | 1992 | 3347 | 15299 | 425 | 137 | 1101 | 1663 | 16962 |
| $CO_2$-eq [mol d$^{-1}$] | 10342 | 2247 | 3753 | 16342 | 425 | 137 | 1101 | 1664 | 18005 |

**4 Discussion**

**4.1 Spatial variability**

Our results show considerable spatial variability of CH$_4$ fluxes from aquatic sites. This confirms earlier observations (Staniek, 2018) and can be explained by the fact that CH$_4$ is primarily produced in the sediment and thus, depends on the spatial heterogeneity of the sediments. Although there was little CH$_4$ production also in the water, our incubation experiment confirms that the sediment was the dominant source of CH$_4$ in River Elbe. Sediment deposition in rivers is highly heterogeneous and depends on hydrodynamics (Henning and Hentschel, 2013). Fine material rich in organic matter preferably settles at low stream velocity, in our case in the groyne fields. Thus, our study confirms the hypothesis that the groyne fields are the major source of CH$_4$ in river Elbe (Bussmann et al., 2022). Consistent with observations of Matousu et al (2019) we did not observe ebulltion, suggesting that ebullition is probably of minor importance in River Elbe. However, we can not exclude that ebullition might contribute to variability of CH$_4$ fluxes in dammed sections of the river, where high CH$_4$ concentrations were observed (Bussmann et al., 2022).

The data also revealed a significant difference between aquatic and terrestrial CH$_4$ fluxes (Wilcox Test, $p<0.05$). CH$_4$ fluxes from dry sediments have rarely been measured, and there is still debate about their significance. A recent global survey indicates that these CH$_4$ fluxes probably cannot be neglected (Paranaíba et al., 2021). However, our results confirm previous observations that CH$_4$ fluxes from dry river sediments are rather small compared to CO$_2$ fluxes both in terms of the carbon balance and the global warming effect (Koschorreck et al., 2022). It has been suggested that there might be local hot spots of CH$_4$ emissions (Marcé et al., 2019), but our measurements did not show any evidence for such hot spots.

We also observed considerable spatial variability in the CO$_2$ concentration in the water. Concentrations were lower in the groyne fields, most probably because of higher photosynthetic activity and higher plankton biomass (Pusch and Fischer, 2006). However, different CO$_2$ concentrations did not translate into spatial differences in CO$_2$ fluxes, because higher CO$_2$ concentrations were accompanied by lower gas transfer velocities. Higher gas transfer velocities at the side of the river were probably caused by higher turbulence generated from flow energy dissipation. This highlights the interacting role of both concentration and gas transfer velocity in shaping spatial patterns of GHG fluxes from rivers.

While spatial differences in concentrations have already been acknowledged (Bussmann et al., 2022) our study is, to our knowledge, the first to show small-scale spatial differences in gas transfer velocities in a river. Measuring k$_{600}$ in rivers is not an easy task, because tracer addition approaches, as typically used in small streams (Hope et al., 2001), cannot be applied. Eddy covariance measurements in rivers are possible (Huotari et al., 2013) but we consider River Elbe to be too small to exclude footprint contamination by the shore areas. Also, the eddy covariance technique integrates over larger areas and is thus not suited to address small scale spatial variability. The floating chamber method is probably the only existing method that can be used in intermediate streams and rivers (Lorke et al., 2015). The spatial resolution of the method depends on the duration of the measurement, because the boat is drifting during the measurement. We minimised measuring time to about 2 minutes to optimise spatial resolution. Thus, at the measured flow velocity of 0.8 m s$^{-1}$, a typical flux measurement spanned a

drift path of about 100m, which might be the limit for the spatial resolution in the direction of the flow. Since we were drifting parallel to the shore, however, the method was well suited to distinguish $k_{600}$ between the middle and side of the river. While higher $k_{600}$ values at the side of the river (where the groynes introduce turbulence) were expected, the high $k_{600}$ in the groyne fields are somehow surprising. This indicates that factors like bottom roughness and flow energy dissipation at the banks had a larger effect on turbulence on k than simply flow velocity (which was higher in the middle of the river). Wind was not an important factor controlling $k_{600}$ values in our study because wind speeds were low (Fig.S7). It can be expected that in larger and winding rivers with variable fetch, wind field heterogeneities further contribute to the spatial variability of $k_{600}$. It is reasonable to assume that spatially variable gas transfer velocities should also be considered for exchange of other gases (e.g. Hg, Rn) or in stream metabolism calculations where k600 is used to quantify oxygen exchange between water and atmosphere (Demars et al., 2015).

Compared to the aquatic sites, the $CO_2$ fluxes from terrestrial sites showed considerable inter-habitat variability. Higher $CO_2$ fluxes from darker (= more muddy) sites compared to sandy sites have been used to scale up $CO_2$ fluxes from dry river sediments using remote sensing (Mallast et al., 2020). They can be explained both by higher organic matter content of the muddy sediments as well as higher sediment moisture which favors microbial $CO_2$ production in the sediment (Keller et al., 2020). However, the observed trend of decreasing $CO_2$ fluxes with increasing sediment moisture (Figure 5) shows that wetter conditions not necessarily result in higher fluxes. $CO_2$ fluxes from muddy sediments obviously result from a complex interplay between organic matter availability and moisture dependent gas transport limitation (Keller et al., 2020).

The sediment in the proximity of terrestrial vegetation showed clearly elevated $CO_2$ fluxes, confirming earlier observations (Bolpagni et al., 2017; Koschorreck et al., 2022). Since we excluded plants from our chambers, these elevated $CO_2$ fluxes are probably caused by root respiration. It is well known that in soils root respiration contributes about 50 % to soil respiration (Hanson et al., 2000). It is clear that at vegetated sites our $CO_2$ fluxes cannot be equated with net ecosystem exchange because the plants were excluded from our chambers. To fully assess the effect of terrestrial plants on river $CO_2$ fluxes, measurements using transparent chambers and considering light conditions as well as plant biomass determinations are necessary. The exclusion of plants from our measurements means that we systematically overestimate $CO_2$ fluxes from vegetated sites in the growing season. However, we argue that this bias might be small in our study when $CO_2$ uptake by the plants during the day was probably largely compensated by higher $CO_2$ fluxes due to plant respiration during the night.

For practical reasons it was not possible to measure at all sites simultaneously (Table 1). Thus, our spatial data may contain also a temporal signal. Chamber measurements were done only during a few hours during the day. This did probably not affect our results for k600 (because of rather constant wind and discharge conditions). $CH_4$ fluxes was also not affected, considering the very limited diurnal change of $CH_4$ concentration. Regarding $CO_2$ emissions one may argue that the diurnal amplitude of the $CO_2$ concentration might differ between sites. For $CO_2$ differences between the middle of the river and the groyne fields can be expected to be lower in the night because sediment driven $CO_2$ production might increase $CO_2$ concentrations in the groyne fields during the night. This would further decrease the already low spatial variability of aquatic $CO_2$ emissions –

supporting our conclusions. Thus, we think that our sampling design gave a realistic picture of spatial variability within our study reach.

Taken together, our results show that spatial differences were especially apparent for $CH_4$. The terrestrial habitats need to be considered for $CO_2$ emissions while they can probably be neglected for ecosystem-scale $CH_4$ emissions from river Elbe.

### 4.2 Temporal variability of $CO_2$ and $CH_4$

Our results confirm that diurnal variability of $CO_2$, which has been shown in streams (Attermeyer et al., 2021; Gómez-Gener et al., 2021) as well as marine systems (Honkanen et al., 2021), can also be relevant in rivers. Interestingly, the shape of the diurnal curve of $CO_2$ emissions differed between habitats, showing that different regulatory mechanisms are at play.

At aquatic sites biological fixation and mineralization of carbon led to sinusoidal diurnal $pCO_2$ variations, with a maximum in the morning and a minimum in the afternoon. This pattern was most likely driven by light since the diurnal temperature

amplitude in the water was below 1.5°C.

The temporal pattern at the muddy site was also most likely driven by the interplay between microalgae primary production and respiration, but the shape of the diurnal $CO_2$ curve differed considerably. These data nicely demonstrate that the same regulatory mechanism (light-dependent balance of photosynthesis and respiration) may result in different diurnal pattern depending on the physical environments. In water, changes in biological activity are buffered by the dissolve inorganic carbon

(DIC) pool in the water, resulting in gradual changes of $CO_2$ concentration during the day. At terrestrial sites, switching photosynthesis on and off changes the sediment from a $CO_2$ sink to a source and back again. Since microbial respiration depends on temperature and temperature fluctuations in the sediment were quite large, it is somewhat surprising that we did not see a pronounced temperature signal in the $CO_2$ flux (as in Koschorreck et al. (2022)). A possible explanation is that the $CO_2$ pool in the pore space buffers the effect of fluctuating respiration. Since the flux of $CO_2$ between the sediment and the

atmosphere is driven by the concentration gradient, this results in rather constant $CO_2$ efflux during the night. During the day this efflux is most probably blocked by photosynthetic uptake of $CO_2$ by benthic microalgae. In a laboratory study with marine sediments, a similar fast switching process between plateau-like $CO_2$ production in the dark and $CO_2$ uptake during the light, was observed (Tang and Kristensen, 2007). Detailed investigations of benthic primary production on exposed marine sediments showed that both linear and plateau relationships were obtained between the fluorescence parameter (relative

electron transport rate [rETR]) and the community-level carbon-fixation rate (Migne et al., 2007). The reason for a "plateau behavior" was the migration of some cells to greater depth in order to avoid too much light. Possible physiological explanations might be the existence of alternative electron sinks (e.g., the Mehler reaction or photorespiration) or limitation by Calvin cycle reactions.

At vegetated sites, the diurnal pattern was probably driven by diurnal fluctuating plant metabolism and root respiration.

However, the diurnal $CO_2$ flux curve was not in phase with the light or temperature curve. This can be explained by a hysteresis

effect caused by the transit time of $CO_2$ from the source of its formation (probably the plant roots) and the sediment surface (Koschorreck et al., 2022; Phillips et al., 2011).

From a previous study we know that rain events can reduce $CO_2$ emission from sandy sites – most probably by blocking sediment pores (Koschorreck et al., 2022). We did observe a small positive effect of the light rain in the first night only at the vegetated site. There was obviously too little rain to significantly affect sediment moisture and $CO_2$ fluxes either at the sandy site (were rain water just seeped or evaporated) and at the muddy site (where sediment was already wet).

Diurnal variability of $CH_4$ fluxes was observed in a study focussing on spatio-temporal variability of GHG fluxes in the Danube Delta (Canning et al., 2021). Elevated $CH_4$ fluxes from a floodplain lake and a channel were attributed to stratification and temporary mixing of the water column. In our case the water column was permanently mixed and diurnal variability was not an issue for $CH_4$ fluxes. Hydrodynamic conditions where rather constant during our measurements and wind speed was very low – suggesting that k600 did not change much temporarily. Furthermore existing literature suggests that in rivers wind speed (which is potentially variable during the day) has a small effect on k compared to hydrodynamic parameters (which are rather stable on the timescale of days) (Huotari et al., 2013; Molodtsov et al., 2022). Methane is produced in deeper sediment layers and, thus, is not affected by light driven changes of redox conditions at the very sediment surface. Methane consumption (oxidation) can occur either at the sediment surface or in the water column (Matoušů et al., 2019). A recent study however, suggests that this process is not influenced by light and thus daily variations (Broman et al., 2023). Even if $CH_4$ oxidation at the sediment surface were affected by phototrophic activity, this would not result in fluctuating fluxes at the water surface because these are buffered by the $CH_4$ pool in the water column.

Thus, we have confirmed our hypothesis that diurnal variability is relevant for $CO_2$ but not for $CH_4$. We also show that the shape of the diurnal $CO_2$ flux curve depends on the habitat. As already acknowledged in the literature, this has important implications for monitoring strategies and upscaling (Gómez-Gener et al., 2021).

### 4.3 Implications for measurement strategy and upscaling

Based on our results, the best monitoring strategy for our river reach should consider spatial variability of $CH_4$ and both spatial and temporal variability of $CO_2$. Applying that optimal approach to our data (Table 4) revealed the dominant role of $CO_2$ (due to low $CH_4$ fluxes) and aquatic habitats (due to their larger area).

It is evident that the exact quantification of habitat areas is crucial. Stream surface areas are typically estimated from empirical relations depending on stream order (Raymond et al., 2012) or remote sensing (Palmer and Ruhi, 2018). Estimating river width from annual mean discharge, according to Raymond et al. (2012), reveals in our case a width of 183m. This is similar to the mean width of our reach of 200 m, which we obtain by dividing the water surface area (Table 2) by reach length (0.96 km). Although we found good agreement between measured widths and those calculated using the equation of Raymond et al. (2012), measured widths should always be used for field studies because of the large scatter in the regression and the logarithmic scale used. Thus, if river width is not explicitly measured, expected errors can become considerably larger than

the standard error. These approaches, however, do not include dry sediment areas. In our case 9 % of the total area was dry, which is somewhat lower than the 26 % estimated from remote sensing data during the extreme drought in 2018 (Mallast et al., 2020). During that drought, dry sediment areas showed clear longitudinal variability along the river depending on topography ranging from 2 % to 40 % of the river area being dry. A straightforward strategy would be to determine dry areas for different discharge scenarios to derive a quantitative relation between dry area, water area, and discharge.

We used our dataset to simulate different monitoring approaches and compared them with the optimal approach (Figure 7). Only sampling the river during the day at the side would result in about 50 % underestimation of the real GHG emissions – mostly because higher $CO_2$ fluxes during the night are not considered. Measuring in the middle of the river or even in all habitats during the day would only slightly improve the result. If both $CO_2$ and $CH_4$ fluxes were measured over a 24 h cycle only along the side of the river, we would slightly overestimate emissions because of high $CH_4$ fluxes along the side. The convenient approach of deploying only a $CO_2$ probe at the side of the river would result in about 7 % underestimation of total $CO_2$-eq emissions from our study reach.

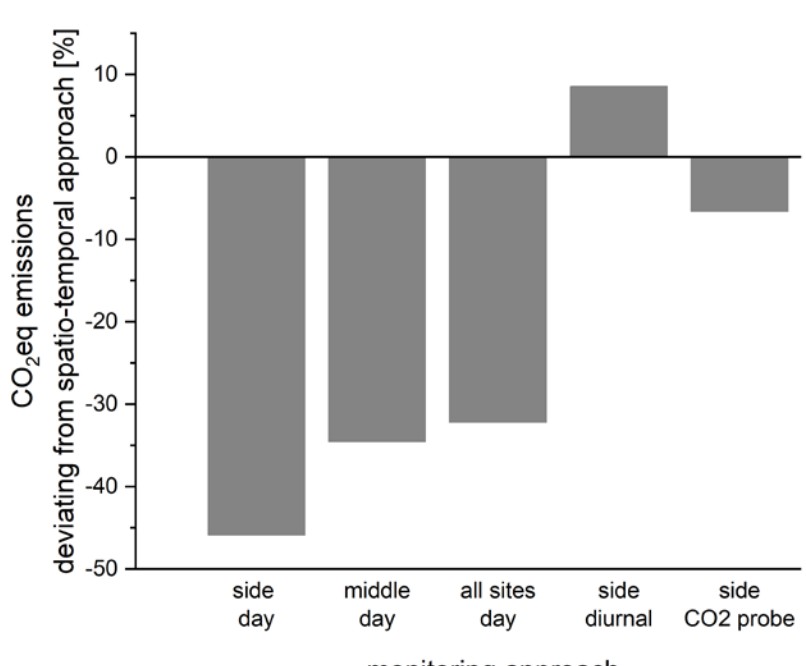

**Figure 7: Deviation of total GHG emissions ($CO_2$-eq) obtained by different monitoring approaches from optimal spatio-temporal approach (spatial variability of $CH_4$ and both spatial and temporal variability of $CO_2$ considered).**

However, these considerations differ depending on the target gas. If we sample only during the day along the side of the river, for example, we would underestimate $CO_2$ emissions by about 50 % but overestimate $CH_4$ emissions by 100 % (Figure S9).

Although our results reveal some general principles, they cannot be simply applied to other systems. For the design of a perfect monitoring strategy for a given river, the particular habitat types and diversity need to be considered. We can also expect that the role of spatial and temporal variability changes with the season, both because habitat areas and regulatory factors like temperature or day-length change (Koschorreck et al., 2022). We would also expect that $CH_4$ variability needs to be re-assessed

if ebullition becomes relevant (Maeck et al., 2014), especially in dammed river sections (Matoušů et al., 2019) or floodplain waters and under warm conditions (Barbosa et al., 2021). More natural river-floodplain systems containing floodplain lakes are known to harbour extreme spatial variability with significant $CH_4$ fluxes (Maier et al., 2021), calling for a more sophisticated monitoring approach (Canning et al., 2021).

**4.4 Conclusion**

Although we only provide a snapshot case study at a German river, we can derive a number of conclusions relevant for the quantification of GHG emission from large temperate rivers.

We show that short term temporal variability is both relevant and complex. It is now evident from several studies that day and night measurements are necessary to come up with realistic emission approaches. $CO_2$ probes are becoming more and more popular. Deploying them in numerous rivers will improve global riverine $CO_2$ emissions estimates. Our results also show that

diurnal pattern may differ between different habitat types. Light and temperature play different roles in shaping temporal variability of $CO_2$ emissions in different habitats.

We also show that spatial variability of $CO_2$ in different aquatic habitats can be considerable but is not the only factor leading to spatially variable fluxes. Also k600 varied between habitats and spatial variability of k600 in rivers cannot be ignored. This point becomes probably less relevant in larger rivers where the side habitat area is small compared to total river area. There is

a need for more studies addressing spatial variability of k600.

We also show principle differences between aquatic and terrestrial GHG emissions both in terms of quantity and regulation. River sediments drying up at low discharge need to be considered at least for $CO_2$ budgets. However, when it comes to total GHG emissions, lower $CH_4$ fluxes compensate for higher $CO_2$ fluxes from dry sediments; this is a scenario already hypothesized for reservoir sediments (Marcé et al., 2019).

Finally, our data show that anthropogenic modification of the river (here: the construction of groynes) has the potential to alter GHG emissions significantly. In our case, the groyne fields nearly doubled $CH_4$ emissions from the river.

**Competing Interests**

The contact author has declared that none of the authors has any competing interests

## Acknowledgements

We would like to thank the captain of RV Albis, S. Bauth and our technician C. Völkner for help during fieldwork. Discharge data were kindly provided by the German Federal Waterways and Shipping Administration (WSV) communicated by the German Federal Institute of Hydrology (BfG). Thanks to Patrick Fink for providing light data and Peifang Leng for commenting on the manuscript. We also thank P. Ronning for proofreading and his valuable comments and to Yao Li for preparing the spatially distibuted ADCP data. This work was supported by funding from the Helmholtz Association in the 520 framework of MOSES (Modular Observation Solutions for Earth Systems) and the POF-4 topic "Coastal Transition Zones under Natural and Human Pressure". The comments of the reviewers significantly improved the manuscript.

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
