# Peer review of "Diurnal versus spatial variability of greenhouse gas emissions from an anthropogenic modified German lowland river"

_Biogeosciences, 2023_

## Author Comment (AC1)

**Reply to reviewer 2**

The work is within the scope of the journal as it addresses CO2 and CH4 pattern in some typical European river. The approach is sound, careful and the arguments are generally convincing

My first major issue – on the real usefulness of diel monitoring of CO2 emissions in rivers.

The sentence in L 25 of the Abstract is questionable. It is unclear why continuous measurements of fCO2 are really necessary given that day/night median values in water habitat (Table 2) which mostly contribute to C emissions (Table 1) are same, within +/- 20% (Table 2). This uncertainty is within the internal measurement uncertainty and hence can be neglected.

We think this argumentation is not valid. As can be seen in Figure 4 diurnal changes of the $CO_2$ flux were phase shifted compared to light. Thus both day and night periods see continuously changing $CO_2$ fluxes resulting in very similar means. It is true that just statistically comparing mean day and night values is not the best approach to analyse these diurnal pattern. If $CO_2$ would only be analyzed once per day, the error is anywhere in the range between 0 and 200 % - the result depends on the shape of the diurnal curve and the sampling time.

We acknowledge that automatic probe measurements are not essential to address this. Maybe sampling at dawn and in the afternoon and then reconstructing a diurnal curve can be sufficient. Thus, probes are very convenient but could be replaced by taking more than one manual measurement during a day. We will modify the discussion at this point.

The second issue is about potential importance of diurnal variations on the annual scale. Peak summer season and sunny weather may not have the same CO2 pattern as cloudy and cooler weather during other period of the year in this part of Germany. At present, extrapolation to year-round time scale is not warranted

We completely agree. We explicitly address short term variability and do not calculate annual budgets. As we state in the manuscript we choose a situation in which we expected large variabilities. We agree to the reviewer that the situation is probably different in other parts of the year. We think, however, that our general conclusions are valid.

Some specific issues

L77-79 Please provide a justification for this hypothesis, based on available literature

We already have this point in the discussion. However, we agree that this needs to be introduced earlier. We will add a section to the introduction explaining why $CH_4$ (which depends on the sediment) is spatially variable.

L201 The rainfall during continuous monitoring is rather unfortunate and highly undesirable event, adding a new dimension (variable). Please explain how it was taken into account.

We do not consider this as unfortunate. Rain is naturally occurring and enabled us to look for the short term effect of rain. The reviewer is right that we did not discuss this point in the

manuscript. From our soil moisture probe data we do not see an effect of rain on soil moisture – most probable because there was so little rain. From another study (Koschorreck et al., 2022) we know that rain can reduce terrestrial $CO_2$ fluxes. However, in our case the rain event was obviously not strong enough to show a visible effect. We will add a sentence to the discussion. We will also add a figure with weather data including rainfall to the SI.

Rain is known to affect also water-atmosphere gas exchange (e.g. (Ho et al., 2000; Ho et al., 1997)). However, we cannot analyze this since our continuous aquatic measurements started after the rain in the first night.

Fig 3 is useful; however, the same plot for pCO2 is needed. Please also consider presenting a plot for CO2 fluxes of this kind to demonstrate spatial variability.

This point was also raised by reviewer 1. The greenhouse gas analyzer could also analyze $CO_2$, however the instrument has never been tested and calibrated for $CO_2$. Thus, no such figure could be provided. We will re-assess our raw data to figure out if $CO_2$ are reliable and can be used for a revision.

L310 Is this 'small' in GHG potential equivalent, or 'small' for the total C balance?

Its both. Given the fact that $CO_2$ fluxes are typically several orders of magnitude larger than $CH_4$ fluxes (in our case more than factor 1000) even multiplying with the GWP does not make a big difference. We will add "both in terms of the carbon balance and the global warming effect".

L315-316 What is the reason of lower Kt at high pCO2 – lower turbulence?

There is not a mechanistic direct link between k and $pCO_2$. What we mean is that in the middle of the river $pCO_2$ was higher but k was lower compared to the sides which resulted in similar flux at all sites.

L353-355 This statement is too general. See works on large tropical and temperate rivers (Mississippi, Congo) or subarctic rivers. On the latter (for instance, Taz, Ket, Lena, see https://doi.org/10.5194/bg-19-1-2022; https://doi.org/10.5194/bg-18-4919-2021; doi: 10.3389/fenvs.2022.98759), the diurnal dynamics of CO2 emissions is not strongly pronounced.

The reviewer questions the general significance of diurnal changes of $CO_2$ emissions from larger rivers. As evidence he provided some references (of which not all seem to be the correct DOI). Vorobyev et al. (2021) indeed did not find pronounced diurnal variability in river Lena, which is not a surprise given the fact that there is little difference between day and night conditions in polar regions in June. There is contrasting evidence regarding the importance of diurnal variability in large tropical rivers (Haque et al., 2022; Ishaque, 1973). Thus we agree that our original statement was too general. We will change "is also relevant" to "can also be relevant".

L368 In water, there is no constant CO2 flux during the night (Fig 5a)

This is true and this is what we write. The sentence in L368 only refers to terrestrial sites.

L372-375 Justification of analogy with marine sediments is necessary

There are several studies on the physiology of benthic algae, mostly on in marine sediments. We see no reason to doubt the assumption that their underlying physiology should be the same for freshwater and marine benthic algae.

**Refernces**

Haque, M.M., Begum, M.S., Nayna, O.K., Tareq, S.M. and Park, J.-H.  2022.  Seasonal shifts in diurnal variations of $pCO_2$ and $O_2$ in the lower Ganges River. Limnology and Oceanography Letters 7(3), 191-201.

Ho, D.T., Asher, W.E., Bliven, L.F., Schlosser, P. and Gordan, E.L.  2000.  On mechanisms of rain-induced air-water gas exchange. J Geophys Res-Oceans 105(C10), 24045-24057.

Ho, D.T., Bliven, L.F., Wanninkhof, R. and Schlosser, P.  1997.  The effect of rain on air-water gas exchange. Tellus Series B-Chemical and Physical Meteorology 49(2), 149-158.

Ishaque, M.  1973.  Intermediates of denitrification in the chemoautotrph Thiobacillus denitrificans. Arch.Microbiol. 94, 269-282.

Koschorreck, M., Knorr, K.H. and Teichert, L.  2022.  Temporal patterns and drivers of $CO_2$ emission from dry sediments in agroyne field of a large river. Biogeosciences 19(22), 5221-5236.

Vorobyev, S.N., Karlsson, J., Kolesnichenko, Y.Y., Korets, M.A. and Pokrovsky, O.S.  2021.  Fluvial carbon dioxide emission from the Lena River basin during the spring flood. Biogeosciences 18(17), 4919-4936.

---

## Author Comment (AC2)

**Reply to reviewer 1**

**Summary of the manuscript**

The reviewed manuscript by Koschorrek et al. quantifies variability of methane and carbon dioxide fluxes between the atmosphere and a temperate low-land river at scales of hours and hundreds of meters. Based on a three-day sampling campaign, including flux chamber measurements in the river and in nearshore areas, the authors found considerable diurnal variability in carbon dioxide fluxes and variability from near-shore to off-shore areas in methane fluxes. The authors also discuss consequences of different sampling strategies for upscaled gas fluxes, concluding that accurate flux estimates require continuous measurements.

**Overall assessment**

Scientific significance: As well introduced by the authors, rivers play an important component of the global carbon cycle and emit carbon gasses at globally significant rates. Yet, there are large uncertainties in emission estimates due to very large spatiotemporal variabilities. The research question on spatiotemporal greenhouse gas fluxes in rivers is not particularly novel, but the focus of this study on small-scale variations (diurnal, near-shore / off-shore) fills a poorly studied niche in the literature that is well worth investigating. I also appreciate the comparison of aquatic and terrestrial gas fluxes, which is rarely done, but highly relevant given that rivers can vary largely in their aerial extend, depending on discharge fluctuations.

Scientific quality: The scope of the study including 3 days of measurements in a 1 km river reach may not appear overly impressive and representative for other conditions. However, relative to many other studies, the authors managed to collect an impressive and interesting data set at very small spatial and temporal scales. Overall, the authors address the research question by using state-of the art techniques. The study design could be acceptable, overall, but some design-related questions should be addressed first (see major concerns below). I agree with most data interpretations and conclusions, but a mismatch in the results shown should be resolved (see major concerns below). I also have a few concerns about the statistical analysis of the data, as outlined below.

Presentation quality: The manuscript is well written, logically structured and clearly and concisely presented. Overall, the figures and tables, including the supplementary material, are adequately chosen and well designed, but I have some concerns and suggestions for improvements, as listed below.

Overall, I find that the manuscript is well within the scope of Biogeosciences.

**Major concerns**

A main focus of the manuscript is to compare spatial and temporal variability in gas fluxes. I wonder to what extent this analysis may be biased by the fact that spatial and temporal assessments were not fully independent? I understand that for practical reasons (limited availability of gas analysers), it is impossible to perform simultaneous measurements at the different locations. However, I would expect a discussion on the consequences of the sampling design for the analysis of spatial and temporal variability in aquatic gas fluxes. For

example, I would like to see at what time the different floating chamber measurements were performed. Given that each measurements takes 2-5 min, I would expect that daytime may affect measurements, in addition to location. Did the authors account for time in their assessment of spatial variability?

The reviewer is right that it is challenging to measure simultaneously at several sites. We partly succeeded here by deploying 3 automatic chambers at different habitats on dry sediments. Thus, at the dry sites we think we adequately addressed spatiotemporal variability simultaneously. The reviewer is right that we did not do so at the aquatic sites. Chamber measurements were done only during a few hours during the day. This will probably not affect our results for k600 (because of rather constant wind and discharge conditions). $CH_4$ fluxes will also not be affected, considering the very limited diurnal change of $CH_4$ concentration. Regarding $CO_2$ emissions one may argue that the diurnal amplitude of the $CO_2$ concentration might differ between sites. For $CO_2$ differences between the middle of the river and the groyne fields can be expected to be lower in the night because sediment driven $CO_2$ production might increase $CO_2$ concentrations in the groyne fields during the night. This would further decrease the already low spatial variability of aquatic $CO_2$ emissions – supporting our conclusions. In a revised manuscript we will add these considerations to the discussion.

Related to the major comment above, it is unclear to me how potential temporal variability in the gas transfer velocity was accounted for in calculations of diel gas fluxes. I appreciate the high temporal resolution of dissolved gas concentrations, but for accurate calculations of gas fluxes, temporal variability in k should also be characterized. K may or may not vary on a diel basis (see e.g. Attermeyer et al. 2021 Comm. Earth&Env). Please clarify how time series fluxes were calculated and discuss any potential shortcomings, in case concentration and k estimates differ in temporal resolution.

We do not expect large differences in K because of rather low constant wind below 4 m/s and discharge. Furthermore existing literature suggests that in rivers wind speed (which is potentially variable during the day) has a small effect on k compared to hydrodynamic parameters (which are rather stable on the timescale of days) (Huotari et al., 2013; Molodtsov et al., 2022). A Figure with windspeed data will be added to the supplement.

I think there is a mismatch in gas fluxes and concentrations shown in Table 1 and in Figures 3/4. According to Table 1, CO2 fluxes range up to 13.9 mmol m-2 h-1, with medians up to 2.8 mmol m-2 h-1. In contrast, the Figure 4 shows maximum fluxes of near 30 mmol m-2 h-1 and medians of up to 10 mmol m-2 h-1. Also, CH4 concentrations in Figure 3 range up to 240 nmol/L, compared to 320 nmol/L in Table 1. Shouldn't the data shown in Table 1 and Figures 3/4 be the same? Table 1 suggests no considerable difference in CO2 fluxes between aquatic and terrestrial habitats, while Figure 4 does. The mismatch may have implications for results (L. 216) and conclusions (L. 432). This issue must be addressed, through corrections or clarifications, before the manuscript can be considered for publication.

In fact the data in Table 1 and Figure 5 are not the same. Table 1 shows the result of manual chamber measurements in the different habitat types including several sites per habitat type. In Figure 4 the temporal data at one site per habitat type (were the probe or automatic chamber was installed) is shown.

However, we discovered an error in table 1 where partly an earlier version of the table was included. The CH4 and CO2 emissions data of the terrestrial sites have the wrong unit

(nmol/m2 s and Mmol/m2 s) and an exponential instead of linear fit was used for flux calculations. Furthermore those data were not complete – a few sites which later were re-classified with respect to habitat type were missing. We also re-assessed the raw data of the chamber measurements and corrected the aquatic CH4 emission data. The correct data (linear fit and unit as in the table column 1) are in the table below. We will correct Table 1 and Figure 4 (see below) as well as Figure 6 in the revision. We are very grateful that the reviewer discovered the wrong numbers.

| | aquatic | | | terrestrial | | |
|---|---|---|---|---|---|---|
| | middle | side | groyne fields | sandy | muddy | vegetated |
| CH$_4$ emissions [$\mu$mol m$^{-2}$ h$^{-1}$] | 12.5 (4-20.8) | 41.7 (21-75) | 39.6 (12.5-54.2) | 0.12 (-0.2-0.38) | 0.03 (-0.83-1.13) | -0.006 (-2.9-0.5) |
| CO$_2$ emissions [mmol m$^{-2}$ h$^{-1}$] | 2.2 (1.86-2.21) | 1.5 (1.07-3.13) | 0.8 (0.21-2.3) | 1.9 (0.7-4.6) | 4.9 (-4.1-16.8) | 11.1 (-2.7– 45.1) |
| CH$_4$ concentration [$\mu$mol L$^{-1}$] | 0.12 (0.11 - 0.16) | 0.18 (0.17 - 0.32) | 0.18 (0.17 -0.21) | - | - | - |
| CO$_2$ concentration [$\mu$mol L$^{-1}$] | 29.3 (28.4-29.6) | 23.4 (15.4-24.2) | 15.7 (15.4-28.8) | - | - | - |
| k600 [m d$^{-1}$] | 2.6 (1.6-5.2) | 5.2 (2.2-10.3) | 4.3 (1.4-5.7) | - | - | - |
| velocity [m s$^{-1}$] | 0.79 (0.72-0.81) | 0.65 (0.22-0.84) | 0.22 (0.09-0.45) | - | - | - |
| total area [m$^2$] | 125,000 | 25,000 | 42,000 | 7,700 | 4,800 | 6,500 |
| total CH$_4$ emissions [mol h$^{-1}$] | 1.6 (0.5 - 2.6) | 1.0 (0.2 - 1.8) | 1.6 (0.5 - 2.3) | 0.009 (-0.002-0.003) | 0.0002 (-0.004-0.005) | -0.00004 (-0.019 - 0.003) |
| total CO$_2$ emissions [mol h$^{-1}$] | 275 (232 - 276) | 37.5 (26.7 - 78.5) | 33.6 (8.8 - 96.6) | 14.3 (5.5-35.4) | 23.6 (-19.3-80.7) | 72.4 (-17 - 293) |

As a result numbers in Table 3 will minimally change to:

| aquatic | | | | terrestrial | | | | all |
|---|---|---|---|---|---|---|---|---|
| middle | side | groyne fields | total | sandy | muddy | vegetated | total | total |

| | | | | | | | | | |
|---|---|---|---|---|---|---|---|---|---|
| $CH_4$ [mol d$^{-1}$] | 37.5 | 25.02 | 40 | 102 | 0.006 | 0.014 | -0.001 | 0 | 91 |
| $CO_2$ [mol d$^{-1}$] | 9960 | 1992 | 3347 | 15299 | 425 | 137 | 1101 | 1663 | 16962 |
| $CO_2$-eq [mol d$^{-1}$] | 10342 | 2247 | 3753 | 16342 | 425 | 137 | 1101 | 1664 | 18005 |

The authors mention the major effect of salty water inflow (river Saale) affecting water chemistry along the western shore (L. 203-206). The authors sampled the western shore and main part of the river, but not the eastern shore, which seems not to be affected by the salty water inflow. I understand that the focus of this study was on the Groynes located along the western shore. However, given the focus on spatial variability of this study, I think it would have been valuable to also study the eastern shore as a "reference" to better evaluate the effect of the Groynes and the salty inflow. Why did the authors did not do any attempt to also study the eastern shore? To what extent could the salty inflow have affected results? Would there be any way to disentangle the spatially overlapping effects of the salty water inflow and the groynes? I would appreciate a brief discussion on this issue.

We did in fact perform chamber measurements on both sides of the river but lumped the data together in the analysis because otherwise our n would be quite low. When looking at the data from both sides separately we do not see large differences. Conductivity differed between sides but the difference was rather small (1100 versus 1300 µS/cm). Thus, we would not expect significant differences in microbial processes at both sides. We think the small conductivity difference as well as the low number of chamber measurements prevents any further analysis of the potential effect of slightly different salt concentrations on GHG emissions. As we write in the manuscript the conductivity difference is a good indicator for limited lateral mixing of the river water.

What is the role of ebullition for gas fluxes in the studied system? Given the potentially large role for total fluxes as well as spatial and temporal variability of methane fluxes, I think this should be discussed more in the manuscript (extending the statement in L. 422). In particular, did you observe sudden jumps in the within-chamber gas measurements that would indicate ebullition? If so, how did you treat such data and how would the exclusion of ebullition affect gas flux estimates?

The reviewer is right that ebullition would be a game-changer for $CH_4$ emissions. We actually did not observe ebullition in our chamber measurements. We cannot fully exclude that ebullition might occur at other sites or times. However, Matousu et al (2019) did also not observe ebullition in the Elbe (with the exception of one harbor). Thus, ebullition does not seem to be very relevant in the Elbe. Although Matousu et al did not observe ebullition in dammed sections of the river we would not exclude ebullition at river sections upstream of weirs (as frequently shown in other studies) and our results might not be valid directly upstream of the only weir in the German part of River Elbe in Geesthacht (where Bussmann et al. 2022 showed elevated $CH_4$ concentrations). We will add a sentence on this to the discussion.

I would like to see more details on the statistical analyses used. For example, the choice of methods described in L. 187-192 should be justified and the used R functions / packages should be explained/cited. What explanatory variables (fixed and random effects) were investigated in the Linear mixed models (L. 191)? Were fluxes always positive so that log-transformation is justified (L. 190)? How was temporal / spatial autocorrelation tested/accounted for in the analyses? How does the correlation analysis and linear mixed

effects modelling help to address the stated research question? Can you please add details of statistical analysis (Wilcox test statistics, mixed effects model parameters / AIC, degrees of freedom). This could be added in the main text or as tables, e.g. in the supplementary material.

We used base-R functions – thus we do not see the need to cite packages. The explanatory variables used in the mixed linear models are mentioned in the results section (l.260-271). We think that having that information in the results part makes the results easier to read.

There were indeed some negative fluxes in our timeseries at the muddy site and a few at the sandy site (see line 246). For those we added the most negative flux as a constant to the data before log calculation. Autocorrelation was inspected visually and choice of variables done based on expert knowledge. We will add that information to the methods section. In the revision we will also add a table with data from the statistical analysis to the supplement.

**Specific comments**

L. 14: Can the authors motivate their statement that most existing studies were carried out in small streams? Perhaps by referring to published work (review, metaanalysis). Personally, I don't have a complete / up-to date overview of the existing literature, but I don't necessarily have the impression that smaller streams are represented more than larger rivers. For air-water gas exchange work in larger rivers, see e.g. Yao et al. (2007, Sci Total Environ), Alin et al. (2011, JGR), Hall et al. (2012, L&O), Beaulieu et al. (2012, JGR), Striegl et al. (2012, GBC), Huotari et al. (2013, GRL), Borges et al. (2016, Nat. Geosci.), Qu et al. (2017, Sci. Reports), Rosentreter et al. (2017, L&O), Paranaiba et al. (2018, ES&T).

We did not perform a robust literature analysis to clarify this point. In the review by Stanley et al. (2016) $CH_4$ concentrations in 652 small to medium size streams compared to 265 in large streams are reported and there are probably much more data on tracer addition derived k values from small streams compared to k values measured with alternative methods in larger systems. However, we agree with the reviewer that the point is probably not so clear and the answer depends a lot on the subject studied. Process studies and studies on the gas transfer velocity are often carried out (for practical reasons) in small streams. On the other hand in global upscaling, larger systems are better represented because the surface area is easier to quantify (Marx et al., 2017). We are aware that there are several studies on river GHG emissions but there are also many studies on smaller streams. In our eyes it does not make much sense to cite a small selection of the existing literature at this point. Since the point is not crucial for our study we would remove the sentence from the abstract.

L. 28 This may be a matter of taste, but could the title "Necessity of upscaling/quantification of GHG emissions from rivers" be shorted? Starting the manuscript with a less bulky title may approach a wider readership.

Good point. We will shorten to "Greenhous gas emission from rivers"

L. 37 Raymond et al. (2013) relied mainly on calculated CO2 based on pH, alkalinity and temperature, not "measured concentrations" as written here.

It was our intention to include both directly and indirectly measured GHG concentrations without going to into detail on how concentrations are derived. We will clarify this to

"…measured in a restricted number of water samples or calculated from other parameters of the carbonate system (pH, alkalinity, and/or DIC)."

L. 38 Perhaps "gas transfer velocities" could be defined/introduced to make the manuscript more accessible for a wider readership?

We agree and will add a sentence explaining "gas transfer velocities".

L. 38 The term "multiplied" confuses me, because the other terms of the equation that is referred to here (concentrations, gas transfer velocity) are simply mentioned without any mathematical characterization of their relationship. I suggest to rephrase the statement to be more consistent in the language.

We will add the actual equation to make this point clear.

L. 38 I agree that most datasets seem to contain weekly or monthly data, but could the authors provide (a) reference(s) for their statement? Perhaps a metaanalysis/review? For example, Marx et al. (2017, Reviews of Geophysics) mentions "knowledge gaps with respect to high-resolution temporal (i.e., diurnal) and spatial variations of carbon fluxes".

Since we talk about typical datasets we think it is maybe more appropriate to cite a few such studies. In the revision we will add a few references.

L. 40/ L. 124 I agree with Lorke et al. (2015, Biogeosciences) that floating chamber measurements can be problematic in flowing water. This has also been evaluated by Vingiani et al. (2021, Biogeosciences) under a range of hydraulic conditions. I would appreciate if the authors could give more details in the methods section on their floating chamber design. How did the authors minimize potential experimental artifacts (e.g. by using "flying" chambers such as described by Lorke et al. and Vingiani et al.)?

We used a "drifting chamber" similar to the ones used in Lorke et al 2015. We will explain our chamber method in more detail in the method section.

L. 53-54 Please provide (a) reference(s) to support the statement "While a single water sample might be representative of a certain specific reach in a small stream this is undoubtedly not the case in larger rivers." Why would spatial variability be higher in larger systems? Greenhouse gas fluxes can be highly heterogeneous in headwater systems (see e.g. Marx et al. 2017, Reviews of Geophysics; Lupon et al. 2019 L&O; Horgby et al. 2019, JGR). I am not aware of any systematic analysis of variability relative to system size, but I would be happy if the authors can substantiate their statement.

This point was also raised by another reviewer. We agree that spatial heterogeneity in rivers is not necessary larger than in small streams. Actually, we consider the comparison of spatial variability on different scales as a very interesting point. In a revised manuscript we will re-formulate our statement in the sense that "less is known on spatial heterogeneity in rivers compared to what is known in streams".

L. 80 Elsewhere in the manuscript it says the campaign was 3 days long, but here it says 4 days. Can you clarify this difference, please?

The duration of our campaign was in fact 4 days but some time was needed for installation and removal of instruments. Thus, time series data comprise up to 3 days of continuous data. We will change it to 3 days here.

L. 91 Why was the outer boundary of the groyne fields set to 15 m into the river? Is this based on previous research?

This is a misunderstanding. As written in our manuscript the outer boundary of the groyne fields were the line between two neighboring groyne heads. We decided to define the side area of the river extending 15m from that line into the river. We indeed had long discussion how to define the side area. We checked flow velocity and water depth but both were not systematically different between the middle of the river and the side. We finally decided to use a fixed distance because this enables easy quantification of areas in the GIS. We choose 15 m because that are approximately 10% of the river width. Visual inspection confirmed that 15m fully included the turbulent areas below the groyne heads.

L. 116 Measurements of turbidity and chlorophyll are mentioned here but there is no data shown. This should be consistent. I would be happy to see data on chlorophyll as it could indicate the level of primary production and hence provide important context to diel CO2 concentrations.

A supplementary figure (S5-b) showing chlorophyll and turbidity will be added.

L. 118 I had to look up the term "moon pool of the albis". I can imagine that there are more potential readers that are unfamiliar with this term. Consider clarification.

We will replace "moon pool" by "ship's duct with direct water supply".

L. 123-174 The authors used three different portable gas analyzers and a gas chromatograph. Have you performed cross-characterizations of the analyzers to make sure that concentration measurements and flux estimates are comparable between the study systems?

Yes. As written in the method section the measurements by the automatic equilibrator were corrected using GC samples. Before deployment the probes were checked in the laboratory by comparing probe readings with GC samples and/or separate measurements using a membrane equilibrator connected to an NDIR analyzer as explained in Koschorreck et al. 2021. The automatic chambers on land were not directly compared to the other analyzers but differences of atmospheric background concentrations measured by the various instruments were quite small (see figure below).

[Figure]

L. 139 Which "instruments" are referred to here?

The degassing unit and greenhouse gas analyzer. Will be specified in the revision.

L. 153 I appreciate that CO2 was measured in the air continuously. However, I cannot find any data on this in the manuscript or any statement on how the data was used. Did you use this data in flux calculations?

Atmospheric $CO_2$ data were indeed used to calculate fluxes from aquatic concentrations. This was necessary because there was a diurnal change of atmospheric $CO_2$ (see figure above). We will add this information to the method section.

L. 158 "Sampling points" are mentioned here, but I would appreciate a clarification of the exact sampling setup, perhaps already in the section with the study site description. How many chambers were deployed in total / per vegetation zone? This is implicit in Figure 1, but it is not clear to me until this point, whether chambers were deployed in all Groyne fields. Also, based on what criteria was the location of the soil flux chamber chosen? Fig. 1c) suggests that the vegetated site C3 was located very close to muddy area, which makes me wonder how representative this site was for the vegetated area?

The sampling points for the chamber were chosen to represent the three different habitats and to assess the temporal and spatial variation. We defined 19 sampling points representing the different habitats (yellow: sandy; green :vegetated, brown: muddy) including the points of the continuous soil flux chambers C1, C2 and C3. Measurements were taken at different times during the campaign in order to cover the diurnal variability. We will include the Figure below in the SI (Figure S9). The typical features of the different habitats are shown in figure S1. Muddy and sandy areas were free from vegetation and could be clearly distinguished from vegetated zones, which were widely covered by typical herbaceous plants such as Persicaria, Inula britannica, and Xanthium strumarium.

[Figure]

L. 179-182 Gas transfer coefficients were calculated from CH4 fluxes and then converted to CO2. This conversion could potentially be erroneous in the presence of bubbles (Klaus et al. 2022, JGR), so I would appreciate a brief note on the role of bubbles in gas exchange in the study system.

As explained above we did not detect any ebullition. But surface bubbles might result from breaking waves. In our case the water surface was rather smooth without breaking waves. We will add this information to the manuscript. We also tried to quantify k600 from our $CO_2$ data. But $CO_2$ concentrations were sometimes close to equilibrium resulting in large uncertainty in $kCO_2$ calculations. That´s why we decided to use CH4 derived k600 also for $CO_2$.

L. 185-186 It is unclear to me why "Probe measurements of CO2 and CH4 concentrations were converted to fluxes using the measured gas transfer velocity of k600 = 5.5 m d-1 (Table 1)." Why was this constant value chosen here, given that it varies substantially, as the data in Table 1 suggests.

We used the k600 value measured at the side of the river since the probes were also installed at the sides. Since we did not measure k600 exactly were the probe was installed and we wanted to be representative for the "site habitat" we used the mean k600 measured in the "side habitat".

L. 186 Please clarify "converted to kCO2 and kCH4 as explained in Striegl et al. (2012)". Do you mean the Schmidt number conversion as explained in L. 182? What is the difference between the conversions you mention in L. 182 and 186?

There was actually no difference. We will replace "as explained in Striegl et al" by "as explained above.

L. 187-188 Please clarify the definition of "day" and "night". Some details are given in Table 2, but they should also appear or be moved to the methods section. Figure 5 suggests some offset between the timing of day-night shifts and changes in PAR. What is the reasoning behind this offset?

Based on sunrise and sunset we defined day as the period between 6:00 and 20:30. We checked our data and found that the PAR data indeed had a different time zone (CET) than the other data (UTC). We will correct Figure 5 accordingly and re-run the statistical analysis. We are sorry and thank you for discovering this.

Figure 3 I appreciate this map, but I wonder why spatial patterns are only shown for CH4 and not for CO2? I would like to see a map of CO2 measurements.

The green house gas analyzer could also analyze $CO_2$, however the instrument has never been tested and calibrated for $CO_2$. Thus, no such figure can be provided. As written in the caption of Table 1 the $CO_2$ concentrations were obtained from GC samples. We will check our raw data and figure out whether the $CO_2$ data of the GHG analyser can be trusted and eventually be used in the paper.

L. 226 What criteria did you use to extract the areas manually from a google earth image? Was there a clear division between the different areas or is the manual extraction prone to uncertainties/errors?

The criteria for the selection of the aquatic habitats was based on the groyne characteristics (groyne heads delimit the groynes, 10% transition area for side habitat, river area for middle habitat). For the terrestrial areas, the corresponding habitats were inferred on the basis of the structures on the land surface and the shade. We agree, there is a slight uncertainty in the manual determination of the areas. In order to minimize the error we based our estimation not only a single image (as written in the text). We derived the estimation from 5 scenes taken during different water level situations in the summertime from 2016 to 2022. In a new version of the manuscript, corresponding uncertainties regarding the area estimation will be included in Table 1.

Figure 4 It is not totally clear to me what the p-values of pair-wise comparisons refer to. Three values are given, but they are all aligned with the same arrow. Please modify the arrows so it becomes clear which p-value belongs to which comparison.

We will improve figure 4 as shown:

[Figure]

L. 235-236 Data on pH and O2 is mentioned here, but I cannot find this data in Figure S5. Please add the data to the figure.

This figure will be added to the supplement as Figure S5c.

Figure 5 Please add units of light and temperature to panel c).

Will be added.

L. 259 / Figure S6 Why did you perform the correlation analysis only for CO2 fluxes, but not for CH4 fluxes?

Because we did not measure $CH_4$ fluxes with the automatic chambers. We will clarify this in the figure legend.

L. 260 Please clarify how you treated the "high" autocorrelation of light and temperature in the statistical analysis.

We did not treat it because both parameters were not part of the final model.

L. 267-268 The statement on fixed and random factors should be moved to the methods section.

We will move that sentence to the methods section.

L. 269 Do you mean the most parsimonious model?

Yes. Will be changed.

L. 279 Why did you not measure temporal changes in CH4 flux at the dry sites?

We did not have an automatic $CH_4$ flux measuring equipment for dry sites. Because $CH_4$ fluxes at the dry sites were very low and previous research suggested very little temporal variability we decided not to do manual measurements in the night. A significant temporal variability of the methane concentration during the survey chamber measurements can be excluded. During the measurements, the continuous chamber in the muddy habitat was used as a base station and was approached and measured several times during a single measurement interval (approx. 2 hours). No temporal trend was detected.

L. 302 and L. 383 Can you provide a reference for the statement that CH4 is primarily produced in the sediment? Methane can potentially also be produced in the water column, even under oxic conditions (e.g. Guenthel et al. 2019, Nat. Comm.). Is anything known about the sources of CH4 in the Elbe river or other lowland rivers?

Yes – $CH_4$ can also be produced in oxic waters but rates are usually much lower than in anoxic sediments. We actually did water and sediment incubations to check for methane production. And yes, the data for methane production in water were much lower ($1.73 \pm 0.46$ pmol/mL/h) than from the sediment slurries ($1.07 \pm 1.69$ nmol/g dry weight/h). We did not add these data to the manuscript because this would blow up the methods section and we felt this information was not essential for the story of the manuscript. We will check whether these data can be included in a revision without making the paper too complicated.

L. 307 I cannot find any statistical support for the statement that there was "a significant difference between aquatic and terrestrial CH4 emissions". Please provide this support (e.g. in Fig. 4).

We will add "(Wilcox Test, p<0.05)" to the text.

L. 319 I appreciate the pioneering effort on small-scale spatial variability in gas transfer velocities. However, the spatial variability is not explicitly shown in the manuscript. Data on k600 is given in Table 1, but only median values and ranges are shown and it remains unclear whether they represent spatial or temporal variability. While showing variability in k600 is not critical to the focus on fluxes in this paper, it might still be interesting to provide more detailed information on spatial vs temporal variability (e.g. CV) in k600 and gas concentrations in the river. I leave it to the authors to decide whether they want to add this information to substantiate the statement in L. 319, or whether they want to leave this part of the story out.

We actually did not measure temporal variability of k600 in our study because hydrodynamic conditions were rather constant during our study period. We will make this clearer in the text. It is actually a weak point of our study that we did not deploy GHG probes at different

locations to cover temporal variability at different sites. We think this is not a serious problem given the rather small differences in concentration between sites. As written in the manuscript to our knowledge our study is the first to look for small scale spatial variability of k in a river. We think it is a good idea to look for spatio-temporal variability of k600 on small scales in future studies.

L. 319-321 The floating chamber is not the only method applicable to rivers. See Huotari et al. 2013 (GRL) for deployment of the eddy-covariance technique in a river. Also, what is a "large stream" relative to a "river"? I suggest to use consistent terms throughout the manuscript.

We agree that eddy covariance can be used in (really) large rivers. However, river Elbe is only 150 m wide which means that the footprint of EC measurements would in most cases be "contaminated" by the river banks. In the study of Huotari et al. for example the river was more than 2 times wider than Elbe. Reliable data could probably only be obtained if the wind would blow exactly parallel to the river. In a river like Elbe this situation is rarely found given the fact that the river is meandering heavily. We will add a sentence discussing this.

Accoring to Vannote et al. (1980) we define lotic waters larger than 6$^{th}$ order (Strahler) as rivers. Since the Elbe has Strahler order 8 we call it a "river". Unfortunately, there is no common and sharp definition of a "large" stream - some scientists (probably used to very large rivers) do not even consider River Elbe to be large. However, River Elbe is among the list of large rivers (length >1.000 km) in Wikipedia, and fulfills the catchment size criterion (148.000 km² is >50.000 km²) (where discharge is dependent on precipitation). We will check the mansucript for consistency avoiding the term "large stream".

L. 326-329 The statement "While higher k600 values at the side of the river were expected," leaves me to wonder why you expected this. The explanation comes indirectly in the following sentence, but perhaps you can rephrase / reorder the sentence to improve logic?

We will rephrase this.

L. 331-333 Why do the authors refer to stream metabolism here? I agree that k is critical to metabolism calculations based on the free-water oxygen method, but there could be many other examples on exchange of other gases (e.g. Hg, Rn) where k values are relevant. Perhaps rephrase the sentence to reflect that metabolism calculations is just one example where k is relevant?

We will rephrase.

L. 342 Why did you exclude plants from the chambers?

This is an interesting point. Existing studies on GHG emissions from dry sediments largely exclude plants – often for practical reasons. This is why in terrestrial studies "ecosystem emissions" are separated from "soil respiration". In case of river sediments we prefer to talk about "sediment GHG emissions" rather than "sediment respiration" because what we directly measure are emissions and these emissions can be affected by other processes than respiration (see Marcé et al 2019). Integrating temporary vegetation on dry sediments into the whole system carbon budget is tricky and would need a completely different approach (e.g. tracing the fate of vegetation after flooding). We will add 1-2 sentences about the complex role of vegetation to the discussion.

We are currently trying to investigate the effect of plants on dry sediments in the dryflux network (https://www.ufz.de/dryflux/).

L. 357 I would replace "obviously" by "most likely", to reflect uncertainties.

We will replace it.

L. 362 Abbreviation "DIC" should be explained.

It´s dissolved inorganic carbon. Will be added.

L. 385 I don't quite understand why the CH4 pool in the water would buffer fluctuations in CH4 emissions caused by CH4 oxidation at the sediment interface. CH4 could also potentially be oxidized in the water column. Some of the co-authors show this for the Elbe river estuary (Matousu et al. 2017, Aqua Sci). Can you please substantiate your argumentation, e.g. by referring to references?

The statement about buffering means that the pool of $CH_4$ dissolved in the water column is not much affected by hypothetical short term changes of sediment-water exchange of $CH_4$. Thus, even if sediment $CH_4$ emissions would change diurnally this would not change much the aquatic $CH_4$ concentration and thus, the $CH_4$ emissions from the water surface.

Regarding $CH_4$ oxidation we will change the text to: "Methane consumption (oxidation) can occur either at the sediment surface or in the water column (Matousu et al 2019). A recent study however, suggests that this process is not influenced by light and thus daily variations (Broman et al., 2023)."

L. 405 / 414 Please use consistent terms to describe the "optimal" or "perfect" approach.

We will unify terms.

Supporting Information: Please clarify the symbols and units of the data included in the excel sheet that contains Time series data. Why are is the spatially resolved data not provided?

We will provide the spatially resolved data in a revision.

Figure S2 I think it would be useful to show a length metric as x-axis.

We will revise the figure accordingly.

Figure S4 Please add letters to the panels. Also, I think that either the panel headings or the the panel descriptions for b and c in the figure caption are mixed up.

Thank you for your recommendation. In the revision we will add the letters to the panels and revise the panel description.

Figure S6 Please give a clarification of the symbols / abbreviations used in the figure.

We will improve the figure legend.

**Technical notes**

We will address all technical points in the revision.

L. 21 imoprove -> improve

L. 54 or those or -> delete "or"

L. 98 The lines in Fig. 1 b are orange, not red as indicated here

L. 187 Here, both "emissions" and "fluxes" are used. I would suggest to use consistent terms throughout the manuscript. Personally, I prefer the more neutral term "fluxes".

L. 259 Remove "tried". It is apparent that you did the analysis.

L. 395 Remove "very"

L. 399 logyrhytmic -> logarithmic

L. 406 under estimation -> underestimation

**References**

Broman, E., Barua, R., Donald, D., Roth, F., Humborg, C., Norkko, A., Jilbert, T., Bonaglia, S. and Nascimento, F.J.A. 2023. No evidence of light inhibition on aerobic methanotrophs in coastal sediments using eDNA and eRNA. Environmental DNA 5(4), 766-781.

Huotari, J., Haapanala, S., Pumpanen, J., Vesala, T. and Ojala, A. 2013. Efficient gas exchange between a boreal river and the atmosphere. Geophys Res Lett 40(21), 5683-5686.

Marx, A., Dusek, J., Jankovec, J., Sanda, M., Vogel, T., van Geldern, R., Hartmann, J. and Barth, J.A.C. 2017. A review of $CO_2$ and associated carbon dynamics in headwater streams: A global perspective. Reviews of Geophysics 55(2), 560-585.

Molodtsov, S., Anis, A., Li, D., Korets, M., Panov, A., Prokushkin, A., Yvon-Lewis, S. and Amon, R.M.W. 2022. Estimation of gas exchange coefficients from observations on the Yenisei River, Russia. Limnol. Oceanogr. Methods 20(12), 781-788.

Stanley, E.H., Casson, N.J., Christel, S.T., Crawford, J.T., Loken, L.C. and Oliver, S.K. 2016. The ecology of methane in streams and rivers: patterns, controls, and global significance. Ecological Monographs 86(2), 146-171.

Vannote, R.L., Minshall, G.W., Cummins, K.W., Sedell, J.R. and Cushing, C.E. 1980. River Continuum Concept. Can. J. Fish. Aquat. Sci. 37(1), 130-137.

---

## Author Comment (AC3)

**Reply to reviewer 3**

Comments to the manuscript by Koschorreck et al, "Diurnal versus spatial variability of greenhouse gas emissions from an anthropogenic modified German lowland river"

**Overview:**

In this study the authors investigate the temporal and spatial variability in greenhouse gas emissions along a German lowland river that has been heavily impacted by anthropogenic activities. Specific focus is on how different greenhouse gases ($CO_2$ and $CH_4$) differ in their respective variability between different locations along a 1 km river reach and over a diurnal temporal scale. The study is based on a three to four days (dependent on variable) long measurement campaign where different locations (middle and side of river channel, different habitats at the terrestrial-aquatic interface) within the river are monitored. The authors conclude that the variability in aquatic emission is gas-specific and that $CH_4$ are more variable on a spatial scale than $CO_2$, whereas $CO_2$ is more variable on the diurnal scale than $CH_4$. They further show that the non-aquatic parts of the river (i.e. parts that are temporally flooded but that were not flooded during the measurement campaign) contributed 10% of the total GHG flux during the campaign.

The manuscript focus on an important topic that is very suitable for publication in Biogeosciences. Although I do not fully agree with the authors that large rivers are much more heterogeneous in their spatial variability of GHG emissions than small streams, I agree that large rivers are somewhat less studied. I also agree that, due to the relative lack of data, GHG emissions from large rivers are often estimated without any validation. Although small streams (< stream order 4) dominate the total global stream and river length, the water surface area that larger rivers (> stream order 4) are representing is a large share of the total surface area of running water. Hence, understanding the temporal and spatial variability and what controls these variabilities are essential for representative GHG emission estimates.

**General comments:**

With this background the manuscript is an important contribution to the research field. I appreciate the detailed and small scale focus of the study which highlight fundamental differences in how $CO_2$ and $CH_4$ are sourced from a central Europe (and likely also elsewhere?) common type of river environment. The manuscript is in general well written and presented but I have some points that needs to be clarified/added prior to a publication of the study. 1) I have some problems to follow the method section and the structure of it. There are also some unclear parts of the methods which at least to me is confusing and which makes it a bit hard to interpret the results.

We will revise the method section

2) The study is based on a variety of different measurement approaches and setups for capturing GHG fluxes at different habitats/scales. It is currently hard to assess the uncertainty of each approach which makes it hard to understand their absolute or relative difference when measurements are compared.

We agree. We will add a table to the methods showing what was measured with which method including method uncertainties.

3) The conclusion section of the study could be stronger given the high detail of the measurements. Currently the text is rather vaguely written and not fully representing, or capitalizing on, the outcome of the study.

The reviewers are a bit contradictory here. On the one hand we are asked to be careful with generalizing from this case-study. But we are also asked to be stronger with our conclusions. We tend to follow the advice to be careful with generalization. However, we will critically assess our conclusions and will formulate them a bit more explicit.

**Detailed comments:**

Ln 15-16, Although not clear but I interpret this sentence starting with "Here quantification…" that small streams (which are mentioned in the sentence before) are not displaying spatial and temporal variability in their GHG emissions. If this is what the authors mean I strongly disagree. I would claim that the spatial and temporal variability in GHG emissions could be even more pronounced in small streams. I however certainly agree with the authors that small scale assessments of temporal and spatial variabilities are rarely made in larger rivers and even less simultaneously. I suggest that the authors rephrase this sentence.

It is actually an interesting question how to compare spatial variability in systems of different size. It was not our intention to address this question and we will rephrase as suggested.

Ln 17, It later on says in the text that the campaign lasted for four days. I suggest that the authors keep it consistent although I think I understand that the difference stems from that different variables/habitats were measured during different number of days.

This point was also raised by another reviewer. We will check the manuscript for consistency.

Ln 24-25, yes the data confirms the hypothesis, but I think it needs to be transparently stated that this was just true for this river section and during the three or four days long campaign. Whether this is a more universal pattern requires measurements covering more extensive temporal and spatial scales.

True – point will be added to the discussion and conclusion.

Ln 29-31, I think the authors can update the CH4 referencing, here and elsewhere suitable, with the very relevant and recent global stream and river studies (Rocher-Ros et al. 2023; Stanley et al. 2023).

We will definetly refer to Rocher-Ros et al. and will check were it makes sense to cite Stanley et al 2023.

Ln 53-54, I think this motivation statement (comparison with small streams) to why it is necessary to study small scale variability in large rivers is not really true and also not really needed. I have measured very high small scale variability along stream channels in both concentrations and emissions of different GHG´s (especially for CH4). In comparison, I have also measured low spatial variability in GHG´s across larger rivers. To conclude, whether concentrations and emissions are spatially variable are highly site specific and not necessary

related to the size of the water body. I agree though that little is known about spatial (and temporal) variability in larger rivers, a good motivation of the study as such. I suggest a rephrasing of this statement.

Thanks for this suggestion. This refocusing of the objectives will improve the story.

Ln 80, in abstract "three days' campaign". I suggest to be consistent.

We will check the manuscript for consistency.

Figure 2, what is meant with "mean low discharge"? Unclear to me.

It is the mean summer discharge during the last 15 years. Will be specified in the revision.

Ln 107, section header. This section contains more than just flow velocity and depth measurements, I suggest to give a more suitable header.

Header will be changed to "Hydrodynamics and basic physicochemical measurements"

Ln 111-113, These velocity measurements are more suitable for the result section to me, or why are they place in the methods?

True. We will move the results of the velocity measurements to the results section.

Ln 117-119, "The water supply for both sensors was the moon pool of the Albis". This comes without any introduction to the reader, what is moon pool and what is Albis?? Please clarify!

Will be rephrased and explained.

Ln 122-147, This section is a bit unclear to me, and at the same time the core of many of the measurements included. I suggest that the authors go through it and make a more logical structure of the text. For example:

- Are the same spatial measurements described starting in the lines 123 and 139? If so, why are they separated in the text? If not, what is the difference between them?

Floating chamber measurements as well as $CO_2$ concentration measurements were completely independently done from $CH_4$ measurements using the degasser. We will add a table to make more clear what was measured with which method and when and were.

- The dissolved gas mapping described in ln 130, why was it just done for CH4 and not also for CO2? I though the LGR instrument handled both gases?

Yes – the LGR can handle both gases. The setup in combination with the gas-equilibrator however, was never been tested for $CO_2$. Thus, we prefered not to use those CO2 data but used discreet water samples analysed by GC for $CO_2$. We will re-assess our raw data and check it the $CO_2$ data of the LGR instrument can be trusted and used for the manuscript.

- The conversion of ppm values of CH4 to concentrations determined from the mapping was made by a regression equation I assume? I suggest to show the data for this conversion in the SI

We will add to the methods section: "The range of concentrations from the water samples used for calibration was rather narrow (178 – 258 nmol/L), thus we used a conversion factor (water sample conc. / ppm from GGA) which was 88.7 ± 23 nM / ppm)."

- Ln 144, again, what is the moon pool of Albis? Maybe obvious for a "ship-based" researcher but not for me.

We will replace "moon pool" by "ship's duct with direct water supply".

- Ln 145-147, how was the CH4 ppm values from the Contros converted to absolute concentrations, similar to above I suggest to show this conversion in some way. As the use of CH4 sensors are in the forefront for this kind of research it would be highly useful for other researcher to show how this was done.

We will add to the methods section: "The relative values of the Contros sensor were converted to concentrations by relating them to water samples measured with a GC, similar to the values from the GGA (LosGatos). The conversion factor here was 0.06 µmol/ppm."

Ln 148, this section contains more than "terrestrial measurements". The first part of the section describes for example atmospheric measurements. I suggest the authors give a more suitable section header.

We will change the header to "Terrestrial and atmospheric measurements".

Ln 185, what probe measurements? The ones conducted at the ship? Please clarify!

We are refering here to the measurments shown in figure 3, performed with the LosGatos on the inflatable boat. We will change the "probe measurements" to "measurements with the transportable GGA (LosGatos)"

Ln 186, why was a fixed value of k600 (5.5 m d-1) used? I don't see 5.5 in Table 1 as referred to. Also, what habitat do these fluxes represent?

We are sorry – that was caused by wrong number in table 1 (see our reply to reviewer 1). The actual k600 for the side habitat (were the probe was installed) was 5.2 m d$^{-1}$. As explained in the answer to the other reviewer, hydrodynamics and wind did not change much during our study. Data in Figure 5b will be corrected accordingly.

Ln 213, I believe it should be "at the sides" instead of "at the sites"? or?

Yes.

Figure 3, why was not CO2 measured at the same time?

See our reply above.

Ln 243, "were" instead of "are".

Yes.

Figure 5, I assume the time point 12.00 on the x-axis refer to mid-day, I suggest to clarify this.

We think it is clear since night is indicated by the grey shaded area.

Table 2 and related text. Here comes one of my larger concerns. It is currently hard to assess how the different method approaches correspond to each other. Although a range is given for all emissions it is hard to understand with what certainty each individual emission determination (done with a different measurement approach) is made with. Some kind of uncertainty estimate would certainly help the reader with this.

We completely agree. In the revision we will estimate the uncertainties of the different approaches and add this information as a table in the methods section.

Figure 6, similar to above, the CV values are good and illustrate well the variability associated with each gas, variability and habitat. However, to what extent these differences in CV values are dependent on variable certainties in the different methods involved in measuring these emissions is currently unknown for the reader.

Same reply as above

Ln 320, The sentence that starts with "The floating chamber…." Used for what? Do you mean GHG emission measurements? What about the eddy covariance method? There are a few examples of the application of the EC method on rivers e.g. Huotari et al. 2013, Guseva et al. 2021. I suggest to rephrase this sentence.

This point was also raised by another reviewer. We will rephrase it and add a sentence about EC measurements and the problem of their footprint in rather small systems.

Ln 390, here refered to as "stream", in other places "river", I suggest to be consistent and use river. Here and elsewhere in this section specifically, but also throughout the ms.

We agree and will change this.

Ln 399, "logyrhymtic scale"??

Will be corrected

Ln 405 and onwards, I appreciate this exercise of simulating different monitoring approaches and what you might miss/capture with a certain approach. This is good information for the reader and it also put some more perspectives on the detailed data basis that this study provides.

Thanks

Ln 426, I suggest the authors rewrite the conclusions to make them stronger and better reflecting the outcome of the study. I appreciate the starting sentence that the study "just" represent a snapshot of one river reach, still I think the conclusions could be more concise, focused and with a stronger message. The study deserves that.

See our reply above. We will critically assess our conclusions and will formulate them a bit more explicit.

**References**

Guseva, S., Aurela, M., Cortés, A., Kivi, R., Lotsari, E., MacIntyre, S., et al. (2021). Variable Physical Drivers of Near-Surface Turbulence in a Regulated River. *Water Resources Research, 57*(11), e2020WR027939.

Huotari, J., Haapanala, S., Pumpanen, J., Vesala, T., & Ojala, A. (2013). Efficient gas exchange between a boreal river and the atmosphere.Geophysical Research Letters, *40*(21), 2013GL057705.

Rocher-Ros, G., Stanley, E.H., Loken, L.C. et al. Global methane emissions from rivers and streams. Nature 621, 530–535 (2023). https://doi.org/10.1038/s41586-023-06344-6

Stanley, E. H., Loken, L. C., Casson, N. J., Oliver, S. K., Sponseller, R. A., et al: (2023) GRiMeDB: the Global River Methane Database of concentrations and fluxes, Earth Syst. Sci. Data, 15, 2879–2926, https://doi.org/10.5194/essd-15-2879-2023.

---

## Author Response (AR1)

**Reply to reviewer 1**

**Summary of the manuscript**

The reviewed manuscript by Koschorrek et al. quantifies variability of methane and carbon dioxide fluxes between the atmosphere and a temperate low-land river at scales of hours and hundreds of meters. Based on a three-day sampling campaign, including flux chamber measurements in the river and in nearshore areas, the authors found considerable diurnal variability in carbon dioxide fluxes and variability from near-shore to off-shore areas in methane fluxes. The authors also discuss consequences of different sampling strategies for upscaled gas fluxes, concluding that accurate flux estimates require continuous measurements.

**Overall assessment**

Scientific significance: As well introduced by the authors, rivers play an important component of the global carbon cycle and emit carbon gasses at globally significant rates. Yet, there are large uncertainties in emission estimates due to very large spatiotemporal variabilities. The research question on spatiotemporal greenhouse gas fluxes in rivers is not particularly novel, but the focus of this study on small-scale variations (diurnal, near-shore / off-shore) fills a poorly studied niche in the literature that is well worth investigating. I also appreciate the comparison of aquatic and terrestrial gas fluxes, which is rarely done, but highly relevant given that rivers can vary largely in their aerial extend, depending on discharge fluctuations.

Scientific quality: The scope of the study including 3 days of measurements in a 1 km river reach may not appear overly impressive and representative for other conditions. However, relative to many other studies, the authors managed to collect an impressive and interesting data set at very small spatial and temporal scales. Overall, the authors address the research question by using state-of the art techniques. The study design could be acceptable, overall, but some design-related questions should be addressed first (see major concerns below). I agree with most data interpretations and conclusions, but a mismatch in the results shown should be resolved (see major concerns below). I also have a few concerns about the statistical analysis of the data, as outlined below.

Presentation quality: The manuscript is well written, logically structured and clearly and concisely presented. Overall, the figures and tables, including the supplementary material, are adequately chosen and well designed, but I have some concerns and suggestions for improvements, as listed below.

Overall, I find that the manuscript is well within the scope of Biogeosciences.

**Major concerns**

A main focus of the manuscript is to compare spatial and temporal variability in gas fluxes. I wonder to what extent this analysis may be biased by the fact that spatial and temporal assessments were not fully independent? I understand that for practical reasons (limited availability of gas analysers), it is impossible to perform simultaneous measurements at the different locations. However, I would expect a discussion on the consequences of the sampling design for the analysis of spatial and temporal variability in aquatic gas fluxes. For

example, I would like to see at what time the different floating chamber measurements were performed. Given that each measurements takes 2-5 min, I would expect that daytime may affect measurements, in addition to location. Did the authors account for time in their assessment of spatial variability?

The reviewer is right that it is challenging to measure simultaneously at several sites. We partly succeeded here by deploying 3 automatic chambers at different habitats on dry sediments. Thus, at the dry sites we think we adequately addressed spatiotemporal variability simultaneously. The reviewer is right that we did not do so at the aquatic sites. To address this issue we added the information when the different data were measured to table 1. We added to the discussion: "For practical reasons it was not possible to measure at all sites simultaneously (Table 1). Thus, our spatial data may contain also a temporal signal. Chamber measurements were done only during a few hours during the day. This did probably not affect our results for k600 (because of rather constant wind and discharge conditions). $CH_4$ fluxes was also not affected, considering the very limited diurnal change of $CH_4$ concentration. Regarding $CO_2$ emissions one may argue that the diurnal amplitude of the $CO_2$ concentration might differ between sites. For $CO_2$ differences between the middle of the river and the groyne fields can be expected to be lower in the night because sediment driven $CO_2$ production might increase $CO_2$ concentrations in the groyne fields during the night. This would further decrease the already low spatial variability of aquatic $CO_2$ emissions – supporting our conclusions. Thus, we think that our sampling design gave a realistic picture of spatial variability within our study reach."

Related to the major comment above, it is unclear to me how potential temporal variability in the gas transfer velocity was accounted for in calculations of diel gas fluxes. I appreciate the high temporal resolution of dissolved gas concentrations, but for accurate calculations of gas fluxes, temporal variability in k should also be characterized. K may or may not vary on a diel basis (see e.g. Attermeyer et al. 2021 Comm. Earth&Env). Please clarify how time series fluxes were calculated and discuss any potential shortcomings, in case concentration and k estimates differ in temporal resolution.

We do not expect large differences in k because of rather low constant wind below 4 m/s and discharge. A Figure with windspeed data was added to the supplement. We also added to the discussion: "Hydrodynamic conditions where rather constant during our measurements and wind speed was very low – suggesting that k600 did not change much temporarily. Furthermore existing literature suggests that in rivers wind speed (which is potentially variable during the day) has a small effect on k compared to hydrodynamic parameters (which are rather stable on the timescale of days) (Molodtsov et al., 2022; Huotari et al., 2013)."

I think there is a mismatch in gas fluxes and concentrations shown in Table 1 and in Figures 3/4. According to Table 1, CO2 fluxes range up to 13.9 mmol m-2 h-1, with medians up to 2.8 mmol m-2 h-1. In contrast, the Figure 4 shows maximum fluxes of near 30 mmol m-2 h-1 and medians of up to 10 mmol m-2 h-1. Also, CH4 concentrations in Figure 3 range up to 240 nmol/L, compared to 320 nmol/L in Table 1. Shouldn't the data shown in Table 1 and Figures 3/4 be the same? Table 1 suggests no considerable difference in CO2 fluxes between aquatic and terrestrial habitats, while Figure 4 does. The mismatch may have implications for results (L. 216) and conclusions (L. 432). This issue must be addressed, through corrections or clarifications, before the manuscript can be considered for publication.

In fact the data in Table 1 and Figure 5 are not the same. Table 1 shows the result of manual chamber measurements in the different habitat types including several sites per habitat type.

In Figure 4 the temporal data at one site per habitat type (were the probe or automatic chamber was installed) is shown.

However, we discovered an error in table 1 where partly an earlier version of the table was included. The $CH_4$ and $CO_2$ emissions data of the terrestrial sites have the wrong unit (nmol/m2 s and Mmol/m2 s) and an exponential instead of linear fit was used for flux calculations. Furthermore those data were not complete – a few sites which later were re-classified with respect to habitat type were missing. We also re-assessed the raw data of the chamber measurements and corrected the aquatic $CH_4$ emission data. We corrected Table 1, Table 3, Figures 4 - 6 and re-run the statistical analysis (minimal changes). We are very grateful that the reviewer discovered the wrong numbers. As a result our message slightly changes because now the spatial variability of aquatic $CO_2$ and $CH_4$ emissions is similar. We added to the results "Surprisingly, spatial variability of both gases was similar in aquatic habitats (CV = 0.5).

The authors mention the major effect of salty water inflow (river Saale) affecting water chemistry along the western shore (L. 203-206). The authors sampled the western shore and main part of the river, but not the eastern shore, which seems not to be affected by the salty water inflow. I understand that the focus of this study was on the Groynes located along the western shore. However, given the focus on spatial variability of this study, I think it would have been valuable to also study the eastern shore as a "reference" to better evaluate the effect of the Groynes and the salty inflow. Why did the authors did not do any attempt to also study the eastern shore? To what extent could the salty inflow have affected results? Would there be any way to disentangle the spatially overlapping effects of the salty water inflow and the groynes? I would appreciate a brief discussion on this issue.

We did in fact perform chamber measurements on both sides of the river but lumped the data together in the analysis because otherwise our n would be quite low. When looking at the data from both sides separately we do not see large differences. Conductivity differed between sides but the difference was rather small (1100 versus 1300 µS/cm). Thus, we would not expect significant differences in microbial processes at both sides. We think the small conductivity difference as well as the low number of chamber measurements prevents any further analysis of the potential effect of slightly different salt concentrations on GHG emissions. As we write in the manuscript the conductivity difference is a good indicator for limited lateral mixing of the river water. We changed the text to "This slight difference in conductivity most probably does not affect microbial GHG production but it indicates limited lateral mixing of river water even over a large distance.".

What is the role of ebullition for gas fluxes in the studied system? Given the potentially large role for total fluxes as well as spatial and temporal variability of methane fluxes, I think this should be discussed more in the manuscript (extending the statement in L. 422). In particular, did you observe sudden jumps in the within-chamber gas measurements that would indicate ebullition? If so, how did you treat such data and how would the exclusion of ebullition affect gas flux estimates?

The reviewer is right that ebullition would be a game-changer for $CH_4$ emissions. We actually did not observe ebullition in our chamber measurements. We cannot fully exclude that ebullition might occur at other sites or times. However, Matousu et al (2019) did also not observe ebullition in the Elbe (with the exception of one harbor). Thus, ebullition does not seem to be very relevant in the Elbe. Although Matousu et al did not observe ebullition in dammed sections of the river we would not exclude ebullition at river sections upstream of

weirs (as frequently shown in other studies) and our results might not be valid directly upstream of the only weir in the German part of River Elbe in Geesthacht (where Bussmann et al. 2022 showed elevated $CH_4$ concentrations). We added a sentence to the results (3.2) and two sentences to the discussion (4.1).

I would like to see more details on the statistical analyses used. For example, the choice of methods described in L. 187-192 should be justified and the used R functions / packages should be explained/cited. What explanatory variables (fixed and random effects) were investigated in the Linear mixed models (L. 191)? Were fluxes always positive so that log-transformation is justified (L. 190)? How was temporal / spatial autocorrelation tested/accounted for in the analyses? How does the correlation analysis and linear mixed effects modelling help to address the stated research question? Can you please add details of statistical analysis (Wilcox test statistics, mixed effects model parameters / AIC, degrees of freedom). This could be added in the main text or as tables, e.g. in the supplementary material.

We used base-R functions – thus we do not see the need to cite packages. The explanatory variables used in the mixed linear models are mentioned in the results section (l.260-271). We think that having that information in the results part makes the results easier to read.

There were indeed some negative fluxes in our timeseries at the muddy site and a few at the sandy site (see line 246). For those we added the most negative flux as a constant to the data before log calculation. Autocorrelation was inspected visually and choice of variables done based on expert knowledge. We added that information to the methods section. We also added a table to the supplement comparing mixed effect linear models of different complexity (Table S2) incl. $R^2$, AIC, and degrees of freedom.

**Specific comments**

L. 14: Can the authors motivate their statement that most existing studies were carried out in small streams? Perhaps by referring to published work (review, metaanalysis). Personally, I don't have a complete / up-to date overview of the existing literature, but I don't necessarily have the impression that smaller streams are represented more than larger rivers. For air-water gas exchange work in larger rivers, see e.g. Yao et al. (2007, Sci Total Environ), Alin et al. (2011, JGR), Hall et al. (2012, L&O), Beaulieu et al. (2012, JGR), Striegl et al. (2012, GBC), Huotari et al. (2013, GRL), Borges et al. (2016, Nat. Geosci.), Qu et al. (2017, Sci. Reports), Rosentreter et al. (2017, L&O), Paranaiba et al. (2018, ES&T).

We did not perform a robust literature analysis to clarify this point. In the review by Stanley et al. (2016) $CH_4$ concentrations in 652 small to medium size streams compared to 265 in large streams are reported and there are probably much more data on tracer addition derived k values from small streams compared to k values measured with alternative methods in larger systems. However, we agree with the reviewer that the point is probably not so clear and the answer depends a lot on the subject studied. Process studies and studies on the gas transfer velocity are often carried out (for practical reasons) in small streams. On the other hand in global upscaling, larger systems are better represented because the surface area is easier to quantify (Marx et al., 2017). We are aware that there are several studies on river GHG emissions but there are also many studies on smaller streams. In our eyes it does not make much sense to cite a small selection of the existing literature at this point. Since the point is not crucial for our study we removed the sentence from the abstract.

L. 28 This may be a matter of taste, but could the title "Necessity of upscaling/quantification of GHG emissions from rivers" be shorted? Starting the manuscript with a less bulky title may approach a wider readership.

Good point. We shortened to "Greenhous gas emission from rivers"

L. 37 Raymond et al. (2013) relied mainly on calculated CO2 based on pH, alkalinity and temperature, not "measured concentrations" as written here.

It was our intention to include both directly and indirectly measured GHG concentrations without going to into detail on how concentrations are derived. We will clarify this to "…measured in a restricted number of water samples or calculated from other parameters of the carbonate system (pH, alkalinity, and/or DIC)."

L. 38 Perhaps "gas transfer velocities" could be defined/introduced to make the manuscript more accessible for a wider readership?

We agree. We added Equation 1 and a sentence to better explain gas transfer velocity: "The gas transfer velocity is a physical parameter describing diffusive gas exchange at the water surface and typically estimated from hydrodynamic parameters like flow velocity, slope and/or bottom roughness (Raymond and Cole, 2001)."

L. 38 The term "multiplied" confuses me, because the other terms of the equation that is referred to here (concentrations, gas transfer velocity) are simply mentioned without any mathematical characterization of their relationship. I suggest to rephrase the statement to be more consistent in the language.

We added the actual equation to make this point clear.

L. 38 I agree that most datasets seem to contain weekly or monthly data, but could the authors provide (a) reference(s) for their statement? Perhaps a metaanalysis/review? For example, Marx et al. (2017, Reviews of Geophysics) mentions "knowledge gaps with respect to high-resolution temporal (i.e., diurnal) and spatial variations of carbon fluxes".

We added (Stanley et al., 2023) as a reference.

L. 40/ L. 124 I agree with Lorke et al. (2015, Biogeosciences) that floating chamber measurements can be problematic in flowing water. This has also been evaluated by Vingiani et al. (2021, Biogeosciences) under a range of hydraulic conditions. I would appreciate if the authors could give more details in the methods section on their floating chamber design. How did the authors minimize potential experimental artifacts (e.g. by using "flying" chambers such as described by Lorke et al. and Vingiani et al.)?

We used a "drifting chamber" identical to the ones used in Lorke et al 2015. We add: We used exactly the same rectangular drifting chamber (area 0.098 m$^2$, height 0.15 m) as at River Bode in Lorke et al. (2015).

L. 53-54 Please provide (a) reference(s) to support the statement "While a single water sample might be representative of a certain specific reach in a small stream this is undoubtedly not the case in larger rivers." Why would spatial variability be higher in larger systems? Greenhouse gas fluxes can be highly heterogeneous in headwater systems (see e.g. Marx et al.

2017, Reviews of Geophysics; Lupon et al. 2019 L&O; Horgby et al. 2019, JGR). I am not aware of any systematic analysis of variability relative to system size, but I would be happy if the authors can substantiate their statement.

This point was also raised by another reviewer. We agree that spatial heterogeneity in rivers is not necessary larger than in small streams. Actually, we consider the comparison of spatial variability on different scales as a very interesting point. Re changed the sentence to "While there are several studies investigating spatial variability in streams, much less is known about spatial variability of GHG emissions from larger rivers.

L. 80 Elsewhere in the manuscript it says the campaign was 3 days long, but here it says 4 days. Can you clarify this difference, please?

The duration of our campaign was in fact 4 days but some time was needed for installation and removal of instruments. Thus, time series data comprise up to 3 days of continuous data. We will change it to 3 days here.

L. 91 Why was the outer boundary of the groyne fields set to 15 m into the river? Is this based on previous research?

This is a misunderstanding. As written in our manuscript the outer boundary of the groyne fields were the line between two neighboring groyne heads. We decided to define the side area of the river extending 15m from that line into the river. We indeed had long discussion how to define the side area. We checked flow velocity and water depth but both were not systematically different between the middle of the river and the side. We finally decided to use a fixed distance because this enables easy quantification of areas in the GIS. We choose 15 m because that are approximately 10% of the river width. Visual inspection confirmed that 15m fully included the turbulent areas below the groyne heads. We added this information to the methods section.

L. 116 Measurements of turbidity and chlorophyll are mentioned here but there is no data shown. This should be consistent. I would be happy to see data on chlorophyll as it could indicate the level of primary production and hence provide important context to diel $CO_2$ concentrations.

A supplementary figure (S6-b) showing chlorophyll and turbidity was added.

L. 118 I had to look up the term "moon pool of the albis". I can imagine that there are more potential readers that are unfamiliar with this term. Consider clarification.

We replaced "moon pool" by "ship's duct with direct water supply".

L. 123-174 The authors used three different portable gas analyzers and a gas chromatograph. Have you performed cross-characterizations of the analyzers to make sure that concentration measurements and flux estimates are comparable between the study systems?

Yes. As written in the method section the measurements by the automatic equilibrator were corrected using GC samples. Before deployment the probes were checked in the laboratory by comparing probe readings with GC samples and/or separate measurements using a membrane equilibrator connected to an NDIR analyzer as explained in Koschorreck et al. 2021. The automatic chambers on land were not directly compared to the other analyzers but differences

of atmospheric background concentrations measured by the various instruments were quite small (see figure below). We added two sentences to the methods section.

L. 139 Which "instruments" are referred to here?

Changed to "degassing unit and GGA".

L. 153 I appreciate that CO2 was measured in the air continuously. However, I cannot find any data on this in the manuscript or any statement on how the data was used. Did you use this data in flux calculations?

Atmospheric $CO_2$ data were indeed used to calculate fluxes from aquatic concentrations. This was necessary because there was a diurnal change of atmospheric $CO_2$. We added this information to the method section.

L. 158 "Sampling points" are mentioned here, but I would appreciate a clarification of the exact sampling setup, perhaps already in the section with the study site description. How many chambers were deployed in total / per vegetation zone? This is implicit in Figure 1, but it is not clear to me until this point, whether chambers were deployed in all Groyne fields. Also, based on what criteria was the location of the soil flux chamber chosen? Fig. 1c) suggests that the vegetated site C3 was located very close to muddy area, which makes me wonder how representative this site was for the vegetated area?

The sampling points for the chamber were chosen to represent the three different habitats and to assess the temporal and spatial variation. We defined 19 sampling points representing the different habitats (yellow: sandy; green :vegetated, brown: muddy) including the points of the continuous soil flux chambers C1, C2 and C3. Measurements were taken at different times during the campaign in order to cover the diurnal variability. We include the Figure below in the SI (Figure S2). The typical features of the different habitats are shown in figure S1. We added to the method section: "To assess spatial variability we measured GHG fluxes at 5 sandy, 5 muddy, and 10 vegetated sites (Figure S2). Muddy and sandy areas were free from vegetation and could be clearly distinguished from vegetated zones, which were widely covered by typical herbaceous plants such as *Persicaria*, *Inula britannica*, and *Xanthium strumarium*."

[Figure]

L. 179-182 Gas transfer coefficients were calculated from CH4 fluxes and then converted to CO2. This conversion could potentially be erroneous in the presence of bubbles (Klaus et al. 2022, JGR), so I would appreciate a brief note on the role of bubbles in gas exchange in the study system.

As explained above we did not detect any ebullition. But surface bubbles might result from breaking waves. As written in the results (3.1) the water surface was rather smooth without larger waves. We also tried to quantify k600 from our $CO_2$ data. But $CO_2$ concentrations were sometimes close to equilibrium resulting in large uncertainty in $kCO_2$ calculations. That´s why we decided to use $CH_4$ derived k600 also for $CO_2$.

L. 185-186 It is unclear to me why "Probe measurements of CO2 and CH4 concentrations were converted to fluxes using the measured gas transfer velocity of k600 = 5.5 m d-1 (Table 1)." Why was this constant value chosen here, given that it varies substantially, as the data in Table 1 suggests.

We used the k600 value measured at the side of the river since the probes were also installed at the sides. Since we did not measure k600 exactly were the probe was installed and we wanted to be representative for the "site habitat" we used the mean k600 measured in the "side habitat".

L. 186 Please clarify "converted to kCO2 and kCH4 as explained in Striegl et al. (2012)". Do you mean the Schmidt number conversion as explained in L. 182? What is the difference between the conversions you mention in L. 182 and 186?

There was actually no difference. We replaced "as explained in Striegl et al" by "as explained above.

L. 187-188 Please clarify the definition of "day" and "night". Some details are given in Table 2, but they should also appear or be moved to the methods section. Figure 5 suggests some offset between the timing of day-night shifts and changes in PAR. What is the reasoning behind this offset?

Based on sunrise and sunset we defined day as the period between 6:00 and 20:30. We checked our data and found that the PAR data indeed had a different time zone (CET) than the other data (UTC). We corrected Figure 5 accordingly and re-run the statistical analysis (no big changes). We are sorry and thank you for discovering this.

Figure 3 I appreciate this map, but I wonder why spatial patterns are only shown for CH4 and not for CO2? I would like to see a map of CO2 measurements.

The green house gas analyzer could also analyze $CO_2$, however the instrument has never been tested and calibrated for $CO_2$. Thus, no such figure can be provided. As written in the caption of Table 1 the $CO_2$ concentrations were obtained from GC samples. We checked our raw-data for continuous $CO_2$ measurements by the LosGatos instruments. However, the degassing unit was set to a high water flow to allow for a $CH_4$ detection at high spatial resolution. As $CH_4$ has a rather low solubility in water this set-up was still sufficiently sensitive for $CH_4$ detection. In contrast, $CO_2$ has a higher solubility in water and thus is expected to take longer time for degassing. Thus, we do not feel comfortable to publish the spatial distribution of dissolved $CO_2$.

L. 226 What criteria did you use to extract the areas manually from a google earth image? Was there a clear division between the different areas or is the manual extraction prone to uncertainties/errors?

The criteria for the selection of the aquatic habitats was based on the groyne characteristics (groyne heads delimit the groynes, 10% transition area for side habitat, river area for middle habitat). For the terrestrial areas, the corresponding habitats were inferred on the basis of the structures on the land surface and the shade. We agree, there is a slight uncertainty in the manual determination of the areas. In order to estimate the error we based our estimation not only a single image (as written in the text). We derived the estimation from 5 scenes taken during different water level situations in the summertime from 2016 to 2022. We added the range of the different areas to Table 1.

Figure 4 It is not totally clear to me what the p-values of pair-wise comparisons refer to. Three values are given, but they are all aligned with the same arrow. Please modify the arrows so it becomes clear which p-value belongs to which comparison.

We improved figure 4 as shown:

[Figure]

L. 235-236 Data on pH and O2 is mentioned here, but I cannot find this data in Figure S5. Please add the data to the figure.

This figure was added to the supplement to Figure S6.

Figure 5 Please add units of light and temperature to panel c).

added

L. 259 / Figure S6 Why did you perform the correlation analysis only for CO2 fluxes, but not for CH4 fluxes?

Because we did not measure $CH_4$ fluxes with the automatic chambers. We clarified this in the figure legend.

L. 260 Please clarify how you treated the "high" autocorrelation of light and temperature in the statistical analysis.

We did not treat it because both parameters were not part of the final model.

L. 267-268 The statement on fixed and random factors should be moved to the methods section.

We moved that sentence to the methods section.

L. 269 Do you mean the most parsimonious model?

Yes. Changed.

L. 279 Why did you not measure temporal changes in CH4 flux at the dry sites?

We did not have an automatic $CH_4$ flux measuring equipment for dry sites. Because $CH_4$ fluxes at the dry sites were very low and previous research suggested very little temporal variability we decided not to do manual measurements in the night. We did not observe any significant temporal variability of the methane concentration and flux during the survey

chamber measurements: during those measurements, the continuous chamber in the muddy habitat was used as a base station and was approached and measured several times. No temporal trend was detected.

L. 302 and L. 383 Can you provide a reference for the statement that CH4 is primarily produced in the sediment? Methane can potentially also be produced in the water column, even under oxic conditions (e.g. Guenthel et al. 2019, Nat. Comm.). Is anything known about the sources of CH4 in the Elbe river or other lowland rivers?

Yes – $CH_4$ can also be produced in oxic waters but rates are usually much lower than in anoxic sediments. We actually did water and sediment incubations to check for methane production. And yes, the data for methane production in water were much lower than from the sediment slurries. We originally did not add these data to the manuscript because this would blow up the methods section and we felt this information was not essential for the story of the manuscript. As a compromise we now added the method description of our sediment incubations to the supplement. We added to the results: "Sediment incubations (methods in SI) confirmed that $CH_4$ was mainly produced in the sediment. In sediment samples from a groyne field $CH_4$ was produced with a rate of $2095 \pm 2781$ mol $L^{-1}$ $h^{-1}$. Surprisingly, oxic water samples also produced methane with a low rate of $1.73 \pm 0.5$ mol $L^{-1}$ $h^{-1}$." And to the discussion: "Although there was little $CH_4$ production also in the water, our incubation experiment confirms that the sediment was the dominant source of $CH_4$ in River Elbe."

L. 307 I cannot find any statistical support for the statement that there was "a significant difference between aquatic and terrestrial CH4 emissions". Please provide this support (e.g. in Fig. 4).

We added "(Wilcox Test, p<0.05)" to the text.

L. 319 I appreciate the pioneering effort on small-scale spatial variability in gas transfer velocities. However, the spatial variability is not explicitly shown in the manuscript. Data on k600 is given in Table 1, but only median values and ranges are shown and it remains unclear whether they represent spatial or temporal variability. While showing variability in k600 is not critical to the focus on fluxes in this paper, it might still be interesting to provide more detailed information on spatial vs temporal variability (e.g. CV) in k600 and gas concentrations in the river. I leave it to the authors to decide whether they want to add this information to substantiate the statement in L. 319, or whether they want to leave this part of the story out.

We actually did not measure temporal variability of k600 in our study because hydrodynamic conditions were rather constant during our study period. Table 1 shows only spatial variability. We added to the discussion: "Hydrodynamic conditions where rather constant during our measurements and wind speed was very low – suggesting that k600 did not change much temporarily. ".

It is actually a weak point of our study that we did not deploy GHG probes at different locations to cover temporal variability at different sites. We think this is not a serious problem given the rather small differences in concentration between sites. As written in the manuscript to our knowledge our study is the first to look for small scale spatial variability of k in a river. We think it is a good idea to look for spatio-temporal variability of k600 on small scales in future studies.

L. 319-321 The floating chamber is not the only method applicable to rivers. See Huotari et al. 2013 (GRL) for deployment of the eddy-covariance technique in a river. Also, what is a "large stream" relative to a "river"? I suggest to use consistent terms throughout the manuscript.

We agree that eddy covariance can be used in (really) large rivers. However, river Elbe is only 150 m wide which means that the footprint of EC measurements would in most cases be "contaminated" by the river banks. In the study of Huotari et al. for example the river was more than 2 times wider than Elbe. Reliable data could probably only be obtained if the wind would blow exactly parallel to the river. In a river like Elbe this situation is rarely found given the fact that the river is meandering heavily. We added to the discussion: "Eddy covariance measurements in rivers are possible (Huotari et al., 2013) but we consider River Elbe to be too small to exclude footprint contamination by the shore areas. Also, the eddy covariance technique integrates over larger areas and is thus not suited to address small scale spatial variability.".

According to Vannote et al. (1980) we define lotic waters larger than $6^{th}$ order (Strahler) as rivers. Since the Elbe has Strahler order 8 we call it a "river". Unfortunately, there is no common and sharp definition of a "large" stream - some scientists (probably used to very large rivers) do not even consider River Elbe to be large. However, River Elbe is among the list of large rivers (length >1.000 km) in Wikipedia, and fulfills the catchment size criterion (148.000 km² is >50.000 km²) (where discharge is dependent on precipitation). We checked the manuscript for consistency avoiding the term "large stream".

L. 326-329 The statement "While higher k600 values at the side of the river were expected," leaves me to wonder why you expected this. The explanation comes indirectly in the following sentence, but perhaps you can rephrase / reorder the sentence to improve logic?

We rephrase to: "While higher $k_{600}$ values at the side of the river (where the groynes introduce turbulence) were expected, the high $k_{600}$ in the groyne fields are somehow surprising."

L. 331-333 Why do the authors refer to stream metabolism here? I agree that k is critical to metabolism calculations based on the free-water oxygen method, but there could be many other examples on exchange of other gases (e.g. Hg, Rn) where k values are relevant. Perhaps rephrase the sentence to reflect that metabolism calculations is just one example where k is relevant?

We rephrased to: "It is reasonable to assume that spatially variable gas transfer velocities should also be considered for exchange of other gases /e.g. Hg, Rn) or in stream metabolism calculations where k600 is used to quantify oxygen exchange between water and atmosphere (Demars et al., 2015).".

L. 342 Why did you exclude plants from the chambers?

This is an interesting point. Existing studies on GHG emissions from dry sediments largely exclude plants – often for practical reasons. This is why in terrestrial studies "ecosystem emissions" are separated from "soil respiration". In case of river sediments we prefer to talk about "sediment GHG emissions" rather than "sediment respiration" because what we directly measure are emissions and these emissions can be affected by other processes than respiration (see Marcé et al 2019). As we write in the discussion integrating temporary vegetation on dry sediments into the whole system carbon budget is tricky and would need a completely

different approach (e.g. transparent chambers, tracing the fate of vegetation after flooding). We are currently trying to investigate the effect of plants on dry sediments in the dryflux network (https://www.ufz.de/dryflux/).

L. 357 I would replace "obviously" by "most likely", to reflect uncertainties.

Replaced.

L. 362 Abbreviation "DIC" should be explained.

It´s dissolved inorganic carbon. Was added.

L. 385 I don't quite understand why the CH4 pool in the water would buffer fluctuations in CH4 emissions caused by CH4 oxidation at the sediment interface. CH4 could also potentially be oxidized in the water column. Some of the co-authors show this for the Elbe river estuary (Matousu et al. 2017, Aqua Sci). Can you please substantiate your argumentation, e.g. by referring to references?

The statement about buffering means that the pool of $CH_4$ dissolved in the water column is not much affected by hypothetical short term changes of sediment-water exchange of $CH_4$. Thus, even if sediment $CH_4$ emissions would change diurnally this would not change much the aquatic $CH_4$ concentration and thus, the $CH_4$ emissions from the water surface. We reformulated to make this more clear: "Even if $CH_4$ oxidation at the sediment surface were affected by phototrophic activity, this would not result in fluctuating fluxes at the water surface because these are buffered by the $CH_4$ pool in the water column."

Regarding $CH_4$ oxidation we added to the discussion: "Methane consumption (oxidation) can occur either at the sediment surface or in the water column (Matousu et al 2019). A recent study however, suggests that this process is not influenced by light and thus daily variations (Broman et al., 2023)."

L. 405 / 414 Please use consistent terms to describe the "optimal" or "perfect" approach.

We unified to "optimal".

Supporting Information: Please clarify the symbols and units of the data included in the excel sheet that contains Time series data. Why are is the spatially resolved data not provided?

The spatially resolved data are published separately at https://doi.org/10.5281/zenodo.10069196. We added that link to the caption of table 2.

Figure S2 I think it would be useful to show a length metric as x-axis.

We revised the figure accordingly.

Figure S4 Please add letters to the panels. Also, I think that either the panel headings or the the panel descriptions for b and c in the figure caption are mixed up.

Thank you for your recommendation. We added letters to the panels and revise the panel description.

Figure S6 Please give a clarification of the symbols / abbreviations used in the figure.

We improved the figure legend.

**Technical notes**

L. 21 imoprove -> improve

corrected

L. 54 or those or -> delete "or"

deleted

L. 98 The lines in Fig. 1 b are orange, not red as indicated here

corrected

L. 187 Here, both "emissions" and "fluxes" are used. I would suggest to use consistent terms throughout the manuscript. Personally, I prefer the more neutral term "fluxes".

We also prefer the term "flux". We unified the terminology in the manuscript by talking about "fluxes" if we refer to measured fluxes. However we kept the "emissions" in the context of upscaling and integrating fluxes of different gases using their global warming potential.

L. 259 Remove "tried". It is apparent that you did the analysis.

corrected

L. 395 Remove "very"

removed

L. 399 logyrhytmic -> logarithmic

corrected

L. 406 under estimation -> underestimation

corrected

**References**

Broman, E., Barua, R., Donald, D., Roth, F., Humborg, C., Norkko, A., Jilbert, T., Bonaglia, S., and Nascimento, F. J. A.: No evidence of light inhibition on aerobic methanotrophs in coastal sediments using eDNA and eRNA, Environmental DNA, 5, 766-781, https://doi.org/10.1002/edn3.441, 2023.

Demars, B. O. L., Thompson, J., and Manson, J. R.: Stream metabolism and the open diel oxygen method: Principles, practice, and perspectives, Limnol Oceanogr-Meth, 13, 356-374, 10.1002/lom3.10030, 2015.

Huotari, J., Haapanala, S., Pumpanen, J., Vesala, T., and Ojala, A.: Efficient gas exchange between a boreal river and the atmosphere, Geophys Res Lett, 40, 5683-5686, https://doi.org/10.1002/2013GL057705, 2013.

Lorke, A., Bodmer, P., Noss, C., Alshboul, Z., Koschorreck, M., Somlai-Haase, C., Bastviken, D., Flury, S., McGinnis, D. F., Maeck, A., Mueller, D., and Premke, K.: Technical note: drifting versus anchored flux chambers for measuring greenhouse gas emissions from running waters, Biogeosciences, 12, 7013-7024, 10.5194/bg-12-7013-2015, 2015.

Marx, A., Dusek, J., Jankovec, J., Sanda, M., Vogel, T., van Geldern, R., Hartmann, J., and Barth, J. A. C.: A review of $CO_2$ and associated carbon dynamics in headwater streams: A global perspective, Reviews of Geophysics, 55, 560-585, https://doi.org/10.1002/2016RG000547, 2017.

Molodtsov, S., Anis, A., Li, D., Korets, M., Panov, A., Prokushkin, A., Yvon-Lewis, S., and Amon, R. M. W.: Estimation of gas exchange coefficients from observations on the Yenisei River, Russia, Limnol. Oceanogr. Methods, 20, 781-788, https://doi.org/10.1002/lom3.10519, 2022.

Raymond, P. A., and Cole, J. J.: Gas exchange in rivers and estuaries: Choosing a gas transfer velocity, Estuaries, 24, 312-317, Doi 10.2307/1352954, 2001.

Stanley, E. H., Casson, N. J., Christel, S. T., Crawford, J. T., Loken, L. C., and Oliver, S. K.: The ecology of methane in streams and rivers: patterns, controls, and global significance, Ecological Monographs, 86, 146-171, 10.1890/15-1027.1, 2016.

Stanley, E. H., Loken, L. C., Casson, N. J., Oliver, S. K., Sponseller, R. A., Wallin, M. B., Zhang, L. W., and Rocher-Ros, G.: GRiMeDB: the Global River Methane Database of concentrations and fluxes, Earth Syst Sci Data, 15, 2879-2926, 10.5194/essd-15-2879-2023, 2023.

Vannote, R. L., Minshall, G. W., Cummins, K. W., Sedell, J. R., and Cushing, C. E.: River Continuum Concept, Can. J. Fish. Aquat. Sci., 37, 130-137, Doi 10.1139/F80-017, 1980.

**Reply to reviewer 2**

The work is within the scope of the journal as it addresses CO2 and CH4 pattern in some typical European river. The approach is sound, careful and the arguments are generally convincing

My first major issue – on the real usefulness of diel monitoring of CO2 emissions in rivers.

The sentence in L 25 of the Abstract is questionable. It is unclear why continuous measurements of fCO2 are really necessary given that day/night median values in water habitat (Table 2) which mostly contribute to C emissions (Table 1) are same, within +/- 20% (Table 2). This uncertainty is within the internal measurement uncertainty and hence can be neglected.

We think that argumentation is not valid. As can be seen in Figure 5 diurnal changes of the $CO_2$ flux were phase shifted compared to light. Thus both day and night periods see continuously changing $CO_2$ fluxes resulting in very similar means. It is true that just statistically comparing mean day and night values is not the best approach to analyze these diurnal pattern. If $CO_2$ would only be analyzed once per day, the error is anywhere in the range between 0 and 200 % - the result depends on the shape of the diurnal curve and the sampling time.

We acknowledge that automatic probe measurements are not essential to address this. Maybe sampling at dawn and in the afternoon and then reconstructing a diurnal curve can be sufficient. Thus, probes are very convenient but could be replaced by taking more than one manual measurement during a day. We modified the abstract: "Continuous measurements or at least sampling at different times of the day are most likely necessary for reliable quantification of river GHG emissions.". In the discussion (4.4) we replaced "continuous" by "day and night".

The second issue is about potential importance of diurnal variations on the annual scale. Peak summer season and sunny weather may not have the same CO2 pattern as cloudy and cooler weather during other period of the year in this part of Germany. At present, extrapolation to year-round time scale is not warranted

We completely agree. We explicitly address short term variability and do not calculate annual budgets. As we state in the manuscript we choose a situation in which we expected large variabilities. We agree to the reviewer that the situation is probably different in other parts of the year. We think, however, that our general conclusions are valid. We added to the discussion: "We can also expect that the role of spatial and temporal variability changes with the season, both because habitat areas and regulatory factors like temperature or day-length change"

Some specific issues

L77-79 Please provide a justification for this hypothesis, based on available literature

We already have this point in the discussion. However, we agree that this needs to be introduced earlier. We added a sentence to the introduction explaining why $CH_4$ (which

depends on the sediment) is spatially variable: "The sediment is the predominant source of $CH_4$ in streams (Stanley et al., 2016) and spatial variability of $CH_4$ production in rivers is known to be controlled by sediment deposition (Maeck et al., 2013)."

L201 The rainfall during continuous monitoring is rather unfortunate and highly undesirable event, adding a new dimension (variable). Please explain how it was taken into account.

We do not consider this as unfortunate. Rain is naturally occurring and enabled us to look for the short term effect of rain. The reviewer is right that we did not discuss this point in the manuscript. We added rainfall data to Figure 5.

From our soil moisture probe data we only see an effect of rain on soil moisture at the vegetated site – most probable because there was so little rain. From another study (Koschorreck et al., 2022) we know that rain can reduce terrestrial $CO_2$ fluxes. However, in our case the rain event had only little effect. We added a sentence to the results: "There was light rain during the first night (Figure 5d) which resulted in a slight increase of sediment moisture as well as $CO_2$ flux only at the vegetated site.". We also added to the discussion: "From a previous study we know that rain events can reduce $CO_2$ emission from sandy sites – most probably by blocking sediment pores (Koschorreck et al., 2022). We did observe a small positive effect of the light rain in the first night only at the vegetated site. There was obviously too little rain to significantly affect sediment moisture and $CO_2$ fluxes either at the sandy site (were rain water just seeped or evaporated) and at the muddy site (where sediment was already wet).".

We add a figure with wind data to the SI and added rainfall data to Figure 5d. Rain is known to affect also water-atmosphere gas exchange (e.g. (Ho et al., 1997; Ho et al., 2000)). However, we cannot analyze this since our continuous aquatic measurements started after the rain in the first night.

Fig 3 is useful; however, the same plot for pCO2 is needed. Please also consider presenting a plot for CO2 fluxes of this kind to demonstrate spatial variability.

This point was also raised by reviewer 1. The greenhouse gas analyzer could also analyze $CO_2$, however the instrument has never been tested and calibrated for $CO_2$. We checked our raw-data for continuous $CO_2$ measurements by the LosGatos instruments. However, the degassing unit was set to a high water flow to allow for a $CH_4$ detection at high spatial resolution. As $CH_4$ has a rather low solubility in water this set-up was still sufficiently sensitive for $CH_4$ detection. In contrast, $CO_2$ has a higher solubility in water and thus is expected to take longer time for degassing. Thus, we do not feel comfortable to publish the spatial distribution of dissolved $CO_2$.

L310 Is this 'small' in GHG potential equivalent, or 'small' for the total C balance?

Its both. Given the fact that $CO_2$ fluxes are typically several orders of magnitude larger than $CH_4$ fluxes (in our case more than factor 1000) even multiplying with the GWP does not make a big difference. We added: "both in terms of the carbon balance and the global warming effect".

L315-316 What is the reason of lower Kt at high pCO2 – lower turbulence?

There is not a mechanistic direct link between k and $pCO_2$. What we mean is that in the middle of the river $pCO_2$ was higher but k was lower compared to the sides which resulted in similar flux at all sites. We added to the discussion: "Higher gas transfer velocities at the side of the river were probably caused by higher turbulence generated from flow energy dissipation."

L353-355 This statement is too general. See works on large tropical and temperate rivers (Mississippi, Congo) or subarctic rivers. On the latter (for instance, Taz, Ket, Lena, see https://doi.org/10.5194/bg-19-1-2022; https://doi.org/10.5194/bg-18-4919-2021; doi: 10.3389/fenvs.2022.98759), the diurnal dynamics of CO2 emissions is not strongly pronounced.

The reviewer questions the general significance of diurnal changes of $CO_2$ emissions from larger rivers. As evidence he provided some references (of which not all seem to be the correct DOI). Vorobyev et al. (2021) indeed did not find pronounced diurnal variability in river Lena, which is not a surprise given the fact that there is little difference between day and night conditions in polar regions in June. There is contrasting evidence regarding the importance of diurnal variability in large tropical rivers (Haque et al., 2022; Ishaque, 1973). Thus we agree that our original statement was too general. We will change "is also relevant" to "can also be relevant". We also add to the introduction: "There is contrasting evidence for the occurrence of diurnal fluctuation of $CO_2$ in larger rivers (Ishaque, 1973; Haque et al., 2022)"

L368 In water, there is no constant CO2 flux during the night (Fig 5a)

This is true and this is what we write. The sentence in L368 only refers to terrestrial sites.

L372-375 Justification of analogy with marine sediments is necessary

There are several studies on the physiology of benthic algae, mostly on in marine sediments. We see no reason to doubt the assumption that their underlying physiology should be the same for freshwater and marine benthic algae.

**References**

Haque, M. M., Begum, M. S., Nayna, O. K., Tareq, S. M., and Park, J.-H.: Seasonal shifts in diurnal variations of $pCO_2$ and $O_2$ in the lower Ganges River, Limnology and Oceanography Letters, 7, 191-201, https://doi.org/10.1002/lol2.10246, 2022.

Ho, D. T., Bliven, L. F., Wanninkhof, R., and Schlosser, P.: The effect of rain on air-water gas exchange, Tellus Series B-Chemical and Physical Meteorology, 49, 149-158, 1997.

Ho, D. T., Asher, W. E., Bliven, L. F., Schlosser, P., and Gordan, E. L.: On mechanisms of rain-induced air-water gas exchange, J Geophys Res-Oceans, 105, 24045-24057, Doi 10.1029/1999jc000280, 2000.

Ishaque, M.: Intermediates of denitrification in the chemoautotrph Thiobacillus denitrificans, Arch.Microbiol., 94, 269-282, 1973.

Koschorreck, M., Knorr, K. H., and Teichert, L.: Temporal patterns and drivers of $CO_2$ emission from dry sediments in agroyne field of a large river, Biogeosciences, 19, 5221-5236, 10.5194/bg-19-5221-2022, 2022.

Maeck, A., DelSontro, T., McGinnis, D. F., Fischer, H., Flury, S., Schmidt, M., Fietzek, P., and Lorke, A.: Sediment Trapping by Dams Creates Methane Emission Hot Spots, Environ. Sci. Technol., 47, 8130-8137, Doi 10.1021/Es4003907, 2013.

Stanley, E. H., Casson, N. J., Christel, S. T., Crawford, J. T., Loken, L. C., and Oliver, S. K.: The ecology of methane in streams and rivers: patterns, controls, and global significance, Ecological Monographs, 86, 146-171, 10.1890/15-1027.1, 2016.

Vorobyev, S. N., Karlsson, J., Kolesnichenko, Y. Y., Korets, M. A., and Pokrovsky, O. S.: Fluvial carbon dioxide emission from the Lena River basin during the spring flood, Biogeosciences, 18, 4919-4936, 10.5194/bg-18-4919-2021, 2021.

**Reply to reviewer 3**

Comments to the manuscript by Koschorreck et al, "Diurnal versus spatial variability of greenhouse gas emissions from an anthropogenic modified German lowland river"

**Overview:**

In this study the authors investigate the temporal and spatial variability in greenhouse gas emissions along a German lowland river that has been heavily impacted by anthropogenic activities. Specific focus is on how different greenhouse gases ($CO_2$ and $CH_4$) differ in their respective variability between different locations along a 1 km river reach and over a diurnal temporal scale. The study is based on a three to four days (dependent on variable) long measurement campaign where different locations (middle and side of river channel, different habitats at the terrestrial-aquatic interface) within the river are monitored. The authors conclude that the variability in aquatic emission is gas-specific and that $CH_4$ are more variable on a spatial scale than $CO_2$, whereas $CO_2$ is more variable on the diurnal scale than $CH_4$. They further show that the non-aquatic parts of the river (i.e. parts that are temporally flooded but that were not flooded during the measurement campaign) contributed 10% of the total GHG flux during the campaign.

The manuscript focus on an important topic that is very suitable for publication in Biogeosciences. Although I do not fully agree with the authors that large rivers are much more heterogeneous in their spatial variability of GHG emissions than small streams, I agree that large rivers are somewhat less studied. I also agree that, due to the relative lack of data, GHG emissions from large rivers are often estimated without any validation. Although small streams (< stream order 4) dominate the total global stream and river length, the water surface area that larger rivers (> stream order 4) are representing is a large share of the total surface area of running water. Hence, understanding the temporal and spatial variability and what controls these variabilities are essential for representative GHG emission estimates.

**General comments:**

With this background the manuscript is an important contribution to the research field. I appreciate the detailed and small scale focus of the study which highlight fundamental differences in how $CO_2$ and $CH_4$ are sourced from a central Europe (and likely also elsewhere?) common type of river environment. The manuscript is in general well written and presented but I have some points that needs to be clarified/added prior to a publication of the study. 1) I have some problems to follow the method section and the structure of it. There are also some unclear parts of the methods which at least to me is confusing and which makes it a bit hard to interpret the results.

We revise the method section as explained below. Especially we added a table showing which approach was used for what.

2) The study is based on a variety of different measurement approaches and setups for capturing GHG fluxes at different habitats/scales. It is currently hard to assess the uncertainty of each approach which makes it hard to understand their absolute or relative difference when measurements are compared.

We agree. We added a table to the methods showing what was measured with which method including method uncertainties.

3) The conclusion section of the study could be stronger given the high detail of the measurements. Currently the text is rather vaguely written and not fully representing, or capitalizing on, the outcome of the study.

The reviewers are a bit contradictory here. On the one hand we are asked to be careful with generalizing from this case-study. But we are also asked to be stronger with our conclusions. We tend to follow the advice to be careful with generalization. We extended the conclusions by adding the probably 2 most predominant new findings of our detailed measurements: "Our results also show that diurnal pattern may differ between different habitat types. Light and temperature play different roles in shaping temporal variability of $CO_2$ emissions in different habitats.

Although there was considerable variability of GHG concentrations in different aquatic habitats, spatial variability of k600 in rivers cannot be ignored. This point becomes probably less relevant in larger rivers where the side habitat area is small compared to total river area. There is a need for more studies addressing spatial variability of k600."

**Detailed comments:**

Ln 15-16, Although not clear but I interpret this sentence starting with "Here quantification…" that small streams (which are mentioned in the sentence before) are not displaying spatial and temporal variability in their GHG emissions. If this is what the authors mean I strongly disagree. I would claim that the spatial and temporal variability in GHG emissions could be even more pronounced in small streams. I however certainly agree with the authors that small scale assessments of temporal and spatial variabilities are rarely made in larger rivers and even less simultaneously. I suggest that the authors rephrase this sentence.

It is actually an interesting question how to compare spatial variability in systems of different size. It was not our intention to address this question. We removed the sentence.

Ln 17, It later on says in the text that the campaign lasted for four days. I suggest that the authors keep it consistent although I think I understand that the difference stems from that different variables/habitats were measured during different number of days.

This point was also raised by another reviewer. We check the manuscript for consistency and changed it to 3 days.

Ln 24-25, yes the data confirms the hypothesis, but I think it needs to be transparently stated that this was just true for this river section and during the three or four days long campaign. Whether this is a more universal pattern requires measurements covering more extensive temporal and spatial scales.

True – we added "from our study reach of River Elbe in summer".

Ln 29-31, I think the authors can update the CH4 referencing, here and elsewhere suitable, with the very relevant and recent global stream and river studies (Rocher-Ros et al. 2023; Stanley et al. 2023).

We added those references.

Ln 53-54, I think this motivation statement (comparison with small streams) to why it is necessary to study small scale variability in large rivers is not really true and also not really needed. I have measured very high small scale variability along stream channels in both concentrations and emissions of different GHG´s (especially for CH4). In comparison, I have also measured low spatial variability in GHG´s across larger rivers. To conclude, whether concentrations and emissions are spatially variable are highly site specific and not necessary related to the size of the water body. I agree though that little is known about spatial (and temporal) variability in larger rivers, a good motivation of the study as such. I suggest a rephrasing of this statement.

Thanks for this suggestion – we agree. A comparison of systems of different size is not our objective. We changed to "While there are several studies investigating spatial variability in streams, much less is known about spatial variability of GHG emissions from larger rivers.".

Ln 80, in abstract "three days' campaign". I suggest to be consistent.

Changed to 3 days.

Figure 2, what is meant with "mean low discharge"? Unclear to me.

This is the "mean low flow discharge", which is the arithmetic mean of the lowest discharge of each month of the year. We changed this notion also in Figure 2 and in the Figure caption.

Ln 107, section header. This section contains more than just flow velocity and depth measurements, I suggest to give a more suitable header.

Header changed to "Hydrodynamics and basic physicochemical measurements"

Ln 111-113, These velocity measurements are more suitable for the result section to me, or why are they place in the methods?

True. We moved the results of the velocity measurements to the results section.

Ln 117-119, "The water supply for both sensors was the moon pool of the Albis". This comes without any introduction to the reader, what is moon pool and what is Albis?? Please clarify!

Was rephrased to "ship's duct with direct water supply".

Ln 122-147, This section is a bit unclear to me, and at the same time the core of many of the measurements included. I suggest that the authors go through it and make a more logical structure of the text. For example:

- Are the same spatial measurements described starting in the lines 123 and 139? If so, why are they separated in the text? If not, what is the difference between them?

Floating chamber measurements as well as $CO_2$ concentration measurements were completely independently done from $CH_4$ measurements using the degasser. We added a table to make more clear what was measured with which method and when.

- The dissolved gas mapping described in ln 130, why was it just done for CH4 and not also for CO2? I though the LGR instrument handled both gases?

Yes – the LGR can handle both gases. The setup in combination with the gas-equilibrator however, was never been tested for $CO_2$. We checked our raw-data of continuous $CO_2$ measurements by the LosGatos instruments. However, the degassing unit was set to a high water flow to allow for a $CH_4$ detection at high spatial resolution. As $CH_4$ has a rather low solubility in water this set-up was still sufficiently sensitive for $CH_4$ detection. In contrast, $CO_2$ has a higher solubility in water and thus is expected to take longer time for degassing. Thus, we do not feel comfortable to publish the spatial distribution of dissolved $CO_2$.

- The conversion of ppm values of CH4 to concentrations determined from the mapping was made by a regression equation I assume? I suggest to show the data for this conversion in the SI

We added to the methods section: "The range of concentrations from the water samples used for calibration was rather narrow (178 – 258 nmol/L), thus we used a conversion factor (water sample conc. / ppm from GGA) which was 88.7 ± 23 nM / ppm)."

- Ln 144, again, what is the moon pool of Albis? Maybe obvious for a "ship-based" researcher but not for me.

We replaced "moon pool" by "ship's duct with direct water supply".

- Ln 145-147, how was the CH4 ppm values from the Contros converted to absolute concentrations, similar to above I suggest to show this conversion in some way. As the use of CH4 sensors are in the forefront for this kind of research it would be highly useful for other researcher to show how this was done.

We added to the methods section: "The ppm values of the Contros sensor were converted to concentrations by relating them to water samples measured with a GC, similar to the values from the GGA (LosGatos). The conversion factor here was 0.06 µmol/L ppm."

Ln 148, this section contains more than "terrestrial measurements". The first part of the section describes for example atmospheric measurements. I suggest the authors give a more suitable section header.

We changed the header to "Terrestrial and atmospheric measurements".

Ln 185, what probe measurements? The ones conducted at the ship? Please clarify!

Yes – the ones at the ship. We modified the sentence to: "Probe measurements of $CO_2$ and $CH_4$ concentrations measured at RV Albis were converted to fluxes using the measured gas transfer velocity of k600 = 5.5 m d$^{-1}$(Table 2).

Ln 186, why was a fixed value of k600 (5.5 m d-1) used? I don't see 5.5 in Table 1 as referred to. Also, what habitat do these fluxes represent?

We are sorry – that was caused by wrong number in table 1 (see our reply to reviewer 1). The actual k600 for the side habitat (were the probe was installed) was 5.2 m d$^{-1}$. As explained in

the answer to the other reviewer, hydrodynamics and wind did not change much during our study.

Ln 213, I believe it should be "at the sides" instead of "at the sites"? or?

Yes.

Figure 3, why was not CO2 measured at the same time?

See our reply above.

Ln 243, "were" instead of "are".

Yes.

Figure 5, I assume the time point 12.00 on the x-axis refer to mid-day, I suggest to clarify this.

We think this is clear since night is indicated by the grey shaded area.

Table 2 and related text. Here comes one of my larger concerns. It is currently hard to assess how the different method approaches correspond to each other. Although a range is given for all emissions it is hard to understand with what certainty each individual emission determination (done with a different measurement approach) is made with. Some kind of uncertainty estimate would certainly help the reader with this.

We completely agree. We estimated the uncertainties of the different approaches and added this information to a new table in the methods section.

Figure 6, similar to above, the CV values are good and illustrate well the variability associated with each gas, variability and habitat. However, to what extent these differences in CV values are dependent on variable certainties in the different methods involved in measuring these emissions is currently unknown for the reader.

Same reply as above

Ln 320, The sentence that starts with "The floating chamber…." Used for what? Do you mean GHG emission measurements? What about the eddy covariance method? There are a few examples of the application of the EC method on rivers e.g. Huotari et al. 2013, Guseva et al. 2021. I suggest to rephrase this sentence.

This point was also raised by another reviewer. We added: "Eddy covariance measurements in rivers are possible (Huotari et al., 2013) but we consider River Elbe to be too small to exclude footprint contamination by the shore areas. Also, the eddy covariance technique integrates over larger areas and is thus not suited to address small scale spatial variability."

Ln 390, here refered to as "stream", in other places "river", I suggest to be consistent and use river. Here and elsewhere in this section specifically, but also throughout the ms.

We agree and changed it to "river".

Ln 399, "logyrhymtic scale"??

corrected

Ln 405 and onwards, I appreciate this exercise of simulating different monitoring approaches and what you might miss/capture with a certain approach. This is good information for the reader and it also put some more perspectives on the detailed data basis that this study provides.

Thanks

Ln 426, I suggest the authors rewrite the conclusions to make them stronger and better reflecting the outcome of the study. I appreciate the starting sentence that the study "just" represent a snapshot of one river reach, still I think the conclusions could be more concise, focused and with a stronger message. The study deserves that.

See our reply above. The conclusions are now more explicit, also addressing spatial variability of k and the different diurnal pattern.

**References**

Guseva, S., Aurela, M., Cortés, A., Kivi, R., Lotsari, E., MacIntyre, S., et al. (2021). Variable Physical Drivers of Near-Surface Turbulence in a Regulated River. *Water Resources Research, 57*(11), e2020WR027939.

Huotari, J., Haapanala, S., Pumpanen, J., Vesala, T., & Ojala, A. (2013). Efficient gas exchange between a boreal river and the atmosphere.Geophysical Research Letters, *40*(21), 2013GL057705.

Rocher-Ros, G., Stanley, E.H., Loken, L.C. et al. Global methane emissions from rivers and streams. Nature 621, 530–535 (2023). https://doi.org/10.1038/s41586-023-06344-6

Stanley, E. H., Loken, L. C., Casson, N. J., Oliver, S. K., Sponseller, R. A., et al: (2023) GRiMeDB: the Global River Methane Database of concentrations and fluxes, Earth Syst. Sci. Data, 15, 2879–2926, https://doi.org/10.5194/essd-15-2879-2023.

---

## Author Response (AR2)

**Reply to reviewers**

**Report #2**

The manuscript has been significantly improved by the authors since the first submission, and I think it will be a highly valuable contribution to the research field. Based on my two major concerns on the initial submission I appreciate 1) the clarification of the different methods used and their different associated uncertainty (i.e. the new Table 1 and related text). It is now much more clear how/when/where the sampling was conducted. Still I miss that the authors do not transparently show the data including regression equations for how the two different sensor derived CH4 data (both spatial mapping and continuously over time on Albis) are converted to "true" concentrations. Although conversion factors and estimated uncertainty are given in the text and table 1, to present the data behind those conversions (in the SI) would improve the strength even further. High-frequent dissolved CH4 data for running water systems are very rare in the literature while such an addition would be very appreciated from the research community.

As outlined in the manuscript text (L 148ff):

„The range of concentrations from the water samples used for calibration was rather narrow (178 – 258 nmol/L), thus we used a conversion factor (water sample conc. / ppm from GGA) which was 88.7 ± 23 nM / ppm)."

Thus, we cannot provide a regression line, but can only provide the basic data as shown in the little table for the SI now:

| Date | Conc. From GC (nmol/l) | from GGA (ppm) | Conversion factor | Mean ± SD |
|------|------------------------|----------------|-------------------|-----------|
| 19.08. 9:14 | 216 | 1.70 | 127.67 | 88.7 ± 23.1 |
| 19.08. 9:49 | 183 | 2.37 | 77.18 | |
| 19.08. 11:04 | 178 | 2.10 | 84.88 | |
| 20.08. 7:37 | 258 | 3.83 | 67.29 | |
| 20.08. 11:22 | 214 | 3.23 | 66.17 | |
| 21.08. 7:20 | 191 | 2.24 | 85.19 | |
| 21.08. 9:06 | 243 | 2.16 | 112.19 | |

I also appreciate 2) that the authors have improved the clarity and strength of the conclusions section to better reflect the study. However, I think that the text flow in those additions could be improved, currently they are added a bit rough and not so well linked to each other. For example:

"Our results also show that diurnal pattern may differ between different habitat types. Light and temperature play different roles in shaping temporal variability of CO2 emissions in different habitats. Although there was considerable variability of GHG concentrations in different aquatic habitats, spatial variability of k600 in rivers cannot be ignored."

We improved the text flow in the conclusions. It reads now: "Although we only provide a snapshot case study at a German river, we can derive a number of conclusions relevant for the quantification of GHG emission from large temperate rivers.

We show that short term temporal variability is both relevant and complex. It is now evident from several studies that day and night measurements are necessary to come up with realistic emission approaches. $CO_2$ probes are becoming more and more popular. Deploying them in numerous rivers

will improve global riverine $CO_2$ emissions estimates. Our results also show that diurnal pattern may differ between different habitat types. Light and temperature play different roles in shaping temporal variability of $CO_2$ emissions in different habitats.

We also show that spatial variability of $CO_2$ in different aquatic habitats can be considerable but is not the only factor leading to spatially variable fluxes. Also k600 varied between habitats and spatial variability of k600 in rivers cannot be ignored. This point becomes probably less relevant in larger rivers where the side habitat area is small compared to total river area. There is a need for more studies addressing spatial variability of k600.

We also show principle differences between aquatic and terrestrial GHG emissions both in terms of quantity and regulation. River sediments drying up at low discharge need to be considered at least for $CO_2$ budgets. However, when it comes to total GHG emissions, lower $CH_4$ fluxes compensate for higher $CO_2$ fluxes from dry sediments; this is a scenario already hypothesized for reservoir sediments (Marcé et al., 2019).

Finally, our data show that anthropogenic modification of the river (here: the construction of groynes) has the potential to alter GHG emissions significantly. In our case, the groyne fields nearly doubled $CH_4$ emissions from the river."

**Report #3**

Overall assessment of the revisions
The authors have revised the manuscript thoroughly and well-grounded and addressed most of my previous comments and the comments by the other reviewers reasonably well. I think that the manuscript will be a valuable contribution for the reasons I outlined earlier in my review. I just have a few minor comments following up on some revisions. My line numbers refer to the version with changes tracked.

Thank you for your careful reading.

Specific comments
Regarding my comment "I think there is a mismatch in gas fluxes and concentrations shown in Table 1 and in Figures 3/4….": thank you for clarifying the difference in the data source between Table 1 and Figures 3/4. To avoid potential confusion among readers, I suggest to clarify this difference in the table / figure captions.

There is already "(data from Figure 3)" in the caption of table 2. We added "(same data as in table 2)" to the caption of Figure 4. We also added "Please note that time series data were measured indepently from spatial data in table 2." To the caption of Figure 5.

Regarding my comment "I would like to see more details on the statistical analyses used…" I am curious to see which base-R function you used for the linear mixed effects modelling. I am not aware of such a base-R function. I am only aware of functions in packages such as nlme or lme4. If package functions were used, they should be properly cited.

We used the lmer function, assuming that this is a base-R function. However,you are right that it is part of the lme4 package. We added the reference "Bates at all 2015). Thank you to pointing on this. We also consider proper citing of R packages important.

L. 226 I am not convinced about the "visual" check of autocorrelation, unless the authors can clarify this further and provide a supporting reference. To my understanding, it is recommended to formally

test if autocorrelation affects the outcome of linear mixed effects models (Zuur, A., Ieno, E. N., Walker, N., Saveliev, A. A., & Smith, G. M. (2009). Mixed effects models and extensions in ecology with R. New York: Springer.). This can be done by comparing models with and without accounting for autocorrelation, e.g. by using the likelihood ratio test.

Thank you for this advice. We performed a linear correlation analysis between light and temperature. We reformulated in the results: "Correlation with temperature or light (which were significantly linearly correlated, F-test $p<0.05$) including all data was not significant (F-test, $p>0.05$)". We did not perform a detailed analysis using likelihood ratio tests since we already used the AIC to detect the influence of the different drivers on model performance and the final model does only contain one driver.

Regarding my comment "L. 185-186 It is unclear to me why "Probe measurements of CO2 and CH4 concentrations were converted to fluxes using the measured gas transfer velocity of k600 = 5.5 m d-1" I think it would be good to clarify in the text that k at the probe location was assumed to be the mean k600 measured in the side habitat.

OK. We added to the methods: "This assumes that k600 at the probe site was equal to the mean k600 measured in the side habitat.

L. 216 Shouldn't it be 5.2 instead of 5.5 m d-1 (as shown in revised Table 2).

Yes – sorry. Typo corrected.

Figure 3: All three reviewers wondered why no map with CO2 data was shown, which makes me think that many readers could be as confused as the reviewers about this decision. I understand and agree with the authors reasoning for not showing the CO2 data. However, I would suggest to add a brief summary of this reasoning as note in the text / figure caption to avoid confusion.

OK. We added to the caption of Fig.3: "$CO_2$ was also measured by the GHG analyzer but data were not used because gas extraction was different for $CH_4$ and $CO_2$ and the system was optimized for $CH_4$."

L. 226-227 The sentence "If autocorrelation between driver variables was visually observed variables were chosen based on expert knowledge." Does not make sense and needs to be rephrased.

We actually only choose variables based on the AIC as written in methods. We removed the sentence.

L. 257 The use of "Surprisingly" is ambiguous here, in my view. Is it surprising that oxic water samples produced methane, or that this production was low? Perhaps just delete "Surprisingly".

We were surprised that the water did produce measurable amounts of $CH_4$ at all. We follow the reviewer and removed the "surprisingly" to avoid any confusion.

L. 304 I think the unit of par contains a typo and should not be divided by 2 ( µmol m-2 s-1/2 )

No – this is not a typo. We divided the PAR data by 2 to fit them into the temperature scale of the figure.

L. 390-391 and L. 415 I don't agree with the statement that wind speeds were "rather constant". Fig. S7 shows a pretty clear diel variability with highs during the day and calm periods during the night. I

agree that wind speeds were low overall as mentioned in L. 390 and I suggest to focus on the low speed and delete "rather constant".

We agree and deleted "rather constant".

Table S1 The Pearson coefficient should not be abbreviated with R2, but with R. Some numbers in the table are negative but a squared number such as R2 cannot be negative.

That is true. The "minus" should indicate that we had a negative correlation. We added that information to the table caption.

Figure 5 The axis labels in the revised Figure 5 are now much smaller than before and very difficult to read. Please increase the font size.
we increased the font size

Technical notes
L. 302 space missing in "weremeasured"

corrected
L. 393 "/" should be "("

corrected
L. 456 "sediemen" should be "sediment"

corrected